# Astrocyte reactivity and inflammation-induced depression-like behaviors are regulated by Orai1 calcium channels

Michaela M. Novakovic[1], Kirill S. Korshunov[1], Rogan A. Grant [2], Megan E. Martin[1], Hiam A. Valencia[2], G. R. Scott Budinger [2], Jelena Radulovic[3] & Murali Prakriya [1,2] ✉

Astrocytes contribute to brain inflammation in neurological disorders but the molecular mechanisms controlling astrocyte reactivity and their relationship to neuroinflammatory endpoints are complex and poorly understood. In this study, we assessed the role of the calcium channel, Orai1, for astrocyte reactivity and inflammation-evoked depression behaviors in mice. Transcriptomics and metabolomics analysis indicated that deletion of Orai1 in astrocytes downregulates genes in inflammation and immunity, metabolism, and cell cycle pathways, and reduces cellular metabolites and ATP production. Systemic inflammation by peripheral lipopolysaccharide (LPS) increases hippocampal inflammatory markers in WT but not in astrocyte Orai1 knockout mice. Loss of Orai1 also blunts inflammation-induced astrocyte $Ca^{2+}$ signaling and inhibitory neurotransmission in the hippocampus. In line with these cellular changes, Orai1 knockout mice showed amelioration of LPS-evoked depression-like behaviors including anhedonia and helplessness. These findings identify Orai1 as an important signaling hub controlling astrocyte reactivity and astrocyte-mediated brain inflammation that is commonly observed in many neurological disorders.

Astrocytes are the most abundant type of glial cell and regulate a wide range of functions in the healthy brain including maintaining optimal ion and neurotransmitter concentrations in the extracellular space, promoting neural and synapse development, and facilitating neuronal metabolism[1]. In some cases, however, astrocytes can also directly contribute to brain damage by undergoing a reactive transformation wherein they lose many of their neurotrophic functions and instead secrete cytokines, chemokines, and growth factors to elevate brain inflammation[1–4]. For example, in depression, a significant global problem with diverse underlying etiologies and mechanisms, astrocytes have been proposed to directly contribute to the observed behavioral changes, and in particular, animal studies have shown that exaggerated astrocyte-driven brain inflammation can lead to depression-like behaviors[5–9]. Despite the explosion of studies implicating astrocytes in neuroinflammation, however, our understanding of the molecular mechanisms and cellular signaling cascades that control astrocyte reactivity and astrocyte-mediated brain inflammation remains poorly understood.

A subtype of depression with clear-cut links to the peripheral immune system is the depression caused by intense peripheral inflammation[10]. In this form of depression, vigorous activation of the peripheral immune system, which can occur during infections or in diseases that induce systemic inflammation such as multiple sclerosis or rheumatoid arthritis, can lead to symptoms of fatigue, anhedonia (inability to experience pleasure), and helplessness[6,7]. These depressed

[1]Department of Pharmacology, Northwestern University Feinberg School of Medicine, Chicago, IL 60611, USA. [2]Department of Medicine, Northwestern University Feinberg School of Medicine, Chicago, IL 60611, USA. [3]Department of Neuroscience, Albert Einstein School of Medicine, Bronx, NY 10461, USA. ✉ e-mail: m-prakriya@northwestern.edu

motivational behaviors are thought to arise, at least in part, from activated immune cells and cytokines including interferon α (IFNα) and tumor necrosis factor-α (TNFα) crossing over into the brain to increase brain inflammation and affect cognitive functions[6,10,11]. Studies have shown that deliberate injection of these and other cytokines in human subjects can induce feelings of anhedonia, fatigue, and helplessness[11]. Astrocytes react rapidly (within hours) to peripheral inflammation and blocking astrocyte activation with broad-spectrum astrocyte antagonists has been shown to ameliorate depression-like behaviors in mice following a peripheral inflammatory stimulus[8,12]. Yet, the specific astrocytic signaling pathways activated by peripheral inflammation are unclear, and in particular, the potential role of $Ca^{2+}$ signaling, which mediates many vital cellular functions in astrocytes[1], has not been directly investigated in the etiology of depression.

As non-excitable cells, astrocyte activity is primarily regulated by elevations in intracellular $Ca^{2+}$ concentration ($[Ca^{2+}]_i$). Stimulation of neurotransmitter or growth factor receptors on astrocytes mobilizes $Ca^{2+}$ from intracellular stores, eliciting $Ca^{2+}$ signals characterized by waves or oscillations[13]. These $Ca^{2+}$ signals play crucial roles in various astrocyte functions, including differentiation, proliferation, gene transcription, and exocytosis[14]. Increasing evidence suggests that aberrant $Ca^{2+}$ signaling in astrocytes not only serves as a hallmark but also functions as a mediator of neurotoxic inflammation[15]. Elevated $Ca^{2+}$ signaling is tightly associated with reactive astrocytes and is observed in response to neuronal hyperexcitability, amyloid plaques, ischemia, and tissue damage[16,17]. Interestingly, studies have shown that inhibiting astrocyte $Ca^{2+}$ signaling in animal models of brain injury or seizures improves disease outcomes[17,18]. However, there are significant gaps in our understanding of how astrocyte $Ca^{2+}$ signaling is linked to neuroinflammation. Which ion channels are involved in astrocyte $Ca^{2+}$ signaling? How are these ion channels coupled to the transcription and secretion of cytokines? What role do they play in driving neuroinflammation? These are important questions as $Ca^{2+}$ channels represent highly promising targets for developing therapeutic interventions to mitigate neurotoxic inflammation in brain diseases.

In many animal cells, a major mechanism for mobilizing $Ca^{2+}$ signaling is store-operated $Ca^{2+}$ entry (SOCE)[19,20]. SOCE is initiated by activation of G-protein coupled receptors (GPCRs) and receptor tyrosine kinases which trigger $Ca^{2+}$ release from the endoplasmic reticulum (ER). The ensuing depletion of ER $Ca^{2+}$ stores is sensed by ER-associated STIM1 and STIM2 proteins, which translocate to ER-plasma membrane junctions to activate $Ca^{2+}$ release-activated $Ca^{2+}$ (CRAC) channels formed by the Orai proteins (Orai1–3)[19] (Supplementary Fig. 1). CRAC channels are particularly well-suited to produce oscillatory and prolonged $Ca^{2+}$ signals that are needed for transcriptional and enzymatic cascades due to their high $Ca^{2+}$ selectivity and low conductance[19]. We and others have previously shown that hippocampal astrocytes express CRAC channels formed by Orai1 and STIM1[21–23], and their activation stimulates agonist-evoked $Ca^{2+}$ signaling and gliotransmitter release via vesicular exocytosis[23]. Orai1 channels are essential for inflammation and immunity mediated by many types of immune cells including microglia[20,24], but their role in regulating astrocyte reactivity is unknown. In this study, we addressed this question using astrocyte-specific Orai1 knockout mice and biochemical, electrophysiological, and behavioral assays. Our results show that Orai1 channels are important regulators of reactive astrogliosis and control the synthesis and release of a wide range of proinflammatory mediators. Moreover, abrogation of Orai1 function mitigates inflammation in the hippocampus in vivo and protects mice against depression-like phenotypes following induction of global inflammation. Together, these results identify Orai1 as a major route for $Ca^{2+}$ signaling in astrocytes and a mediator of the reactive astrocyte phenotype that drives brain inflammation.

## Results

### Orai1 mediates SOCE in hippocampal astrocytes

To address the role of Orai1 channels for astrocyte-mediated brain inflammation and mouse behavior, we generated conditional astrocyte-specific knockout (cKO) mice. We focused our cellular studies on the hippocampus as astrocytes are abundant here and show extensive end-feet coverage of neurons and synapses[25]. Additionally, Orai1 channels play an important role in agonist-evoked hippocampal astrocyte $Ca^{2+}$ signaling and gliotransmitter release[23], suggesting that alterations in hippocampal astrocyte Orai1 signaling may have a broad impact on many effector functions.

We used two types of astrocyte-specific Orai1 knockout (KO) mouse lines for our mechanistic and behavioral studies: (i) For the mechanistic studies involving in vitro analysis of Orai1 function in primary cultured mouse astrocytes and in situ analysis in brain slices, we used Orai1 KO mice generated by crossing Orai1$^{fl/fl}$ mice with the GFAP-Cre deleter strain. As described previously[23], hippocampal astrocytes from Orai1$^{fl/fl\ GFAP-Cre}$ mice show constitutive loss of astrocyte Orai1 expression (Fig. 1a, b, Supplementary Fig. 2a) and store-operated $Ca^{2+}$ entry (Supplementary Fig. 2b, c). Moreover, there is no change in expression of the other Orai or STIM isoforms (Orai2/3 and STIM1/2) (Fig. 1b), confirming that the deletion of Orai1 is selective and not compensated by the other SOCE molecules. (ii) For the in vivo studies examining astrocyte Orai1 channel contributions to neuroinflammation and behavior, we used Orai1 KO mice obtained by crossing Orai1$^{fl/fl}$ mice with the inducible Cre line Aldh1l1-Cre/ERT2[26] and intraperitoneally injected adult (~8 weeks of age) Orai1$^{fl/fl\ Aldh1ll-Cre/ERT2}$ mice with tamoxifen for Orai1 deletion. For both strategies, to analyze Orai1 contributions to SOCE and effector functions in astrocytes, we cultured hippocampal astrocytes from P2-P5 mice using the serum-free AWESAM protocol that yields astrocytes with stellate morphology, long processes, and a more in vivo–like transcriptome than traditional astrocyte cultures[27]. Immunohistochemical analysis of GFAP and Orai1 co-expressing cells in hippocampal brain sections indicated that astrocytes in adult Orai1$^{fl/fl\ Aldh1ll-Cre/ERT2}$ mice treated with tamoxifen showed significant loss of Orai1 staining in astrocytes compared to Orai1$^{fl/fl}$ mice treated with tamoxifen (Supplementary Fig. 3). In contrast to astrocytes, Orai1 expression was unaffected in microglia from Orai1$^{fl/fl\ Aldh1ll-Cre/ERT2}$ mice (Fig. 1c), indicating that loss of Orai1 in this line is selective for astrocytes. These astrocyte-specific conditional Orai1 KO cells and tamoxifen-injected mice will henceforth be referred to as Orai1 cKO cells/mice. Orai1$^{fl/fl}$ mice and astrocytes treated with tamoxifen (referred to as WT mice/cells) were used as controls for the cellular and behavioral tests.

To examine the contribution of Orai1 channels for SOCE, we loaded astrocytes with the $Ca^{2+}$ indicator, fura-2, and measured the entry of $Ca^{2+}$ into the cells following depletion of intracellular $Ca^{2+}$ stores with $1\ \mu M$ thapsigargin (TG) (Fig. 1d–f, Supplementary Fig. 2b, c). Following store depletion with TG, re-addition of extracellular $Ca^{2+}$ (2 mM) resulted in robust SOCE in WT cells which was blocked by a low dose of $La^{3+}$ (2 $\mu M$) (Fig. 1d). By contrast, SOCE was largely abrogated in astrocytes treated with tamoxifen from Orai1$^{fl/fl\ Aldh1ll-Cre/ERT2}$ mice (Fig. 1d–f) and in astrocytes from Orai1$^{fl/fl\ GFAP-Cre}$ mice in which Orai1 is constitutively deleted (Supplementary Fig. 2b, c). Likewise, SOCE induced by thrombin, a PAR agonist that stimulates diverse effects in the brain including coagulation, cytokine production, and inflammation[28], was abolished in Orai1 cKO astrocytes (Supplementary Fig. 2d–f). These results corroborate previous findings indicating that Orai1 is essential for mediating SOCE in mouse astrocytes[22,23]. Note that our studies do not rule out a potential role of Orai2 or Orai3 proteins in SOCE, but rather indicate that Orai1 has an essential, indispensable role in mediating SOCE in mouse astrocytes.

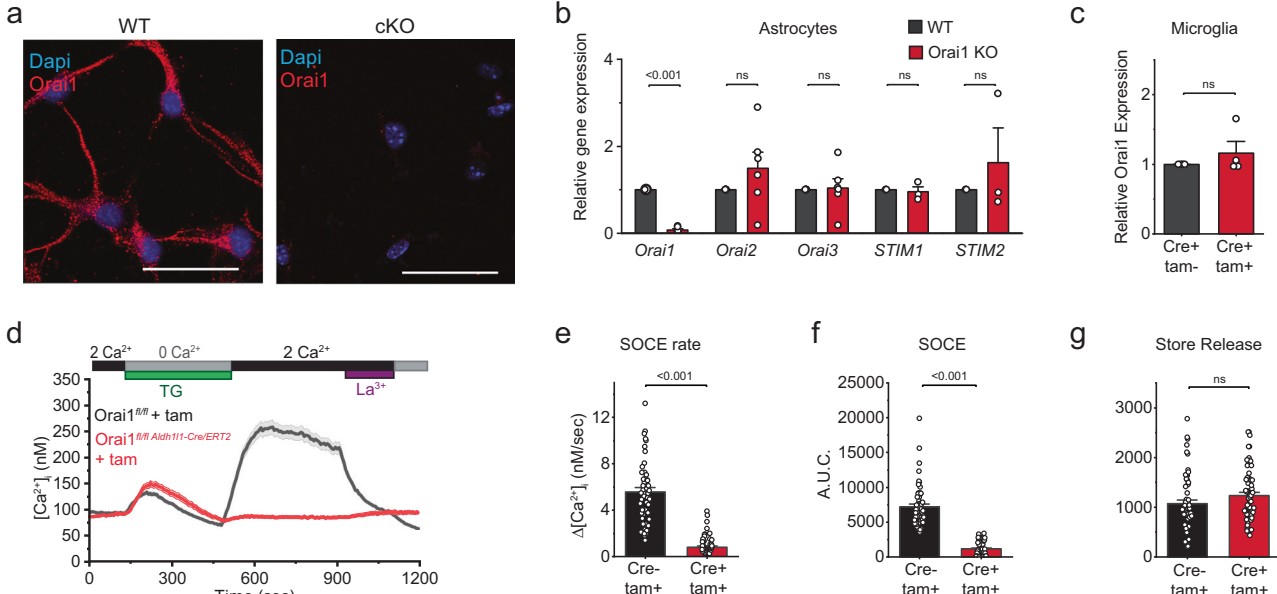

**Fig. 1 | Orai1 mediates SOCE in astrocytes. a** Immunohistochemistry of WT hippocampal astrocytes in culture showing expression of Orai1 in the soma and processes. Orai1 labeling (with a monoclonal antibody, Abcam, M266.1) is lost in astrocytes from Orai1 KO (Orai1$^{fl/fl\ GFAP-Cre}$ mice). Scale bar: 25 μm. **b** Quantification of Orai(1-3) and STIM(1-2) mRNAs via two-step qPCR in WT and Orai1$^{fl/fl\ GFAP-Cre}$ astrocytes. The expression of Orai2-3 and STIM1-2 isoforms is not altered in Orai1 cKO astrocytes. Data are normalized to GAPDH and further normalized to the expression of the corresponding WT mRNA in each case. Mean +/-SEM, $n = 6$ mice for each Orai isoform and $n = 3$ mice for each STIM isoform. **c** Orai1 mRNA is unaffected in microglia from Orai1$^{fl/fl\ Aldh1ll-Cre/ERT2}$ mice. $n = 4$ cultures from 2 mice/group. **d–f** SOCE is abolished in Orai1$^{fl/fl\ Aldh1ll-Cre/ERT2}$ astrocytes exposed to 4-OH tamoxifen (1 μM). SOCE was induced by depleting ER Ca$^{2+}$ stores with thapsigargin (TG, 1 μM)

administered in a 0 Ca$^{2+}$ Ringer's solution. Re-addition of extracellular Ca$^{2+}$ (2 mM) reveals SOCE in WT but not in Orai1$^{fl/fl\ Aldh1ll-Cre/ERT2}$ astrocytes. SOCE in WT astrocytes is also blocked by La$^{3+}$ (2 μM). Panel *E* summarizes the rate of Ca$^{2+}$ influx (nM/sec) following re-addition of extracellular Ca$^{2+}$ following store depletion. Panel *F* summarizes the total amount of calcium over baseline entering the cell for five minutes after Ca$^{2+}$ re-addition. **g** Ca$^{2+}$ release from stores is unaffected in Orai1$^{fl/fl\ Aldh1ll-Cre/ERT2}$ astrocytes. The area under the curve (A.U.C.) indicates the total amount of calcium over baseline released for the three minutes following TG application in 0 mM Ca$^{2+}$. Data were analyzed from the same cells for panels *E-G*. $n = 56$ cells from 3 WT mice and 66 cells from 3 Orai1 cKO mice. Data are presented as mean values +/- SEM. The sex of the pups was not determined. Statistical tests were conducted by two-tailed, unpaired T-tests. Source data are provided as a Source Data file.

## RNA sequencing analysis reveals that Orai1 promotes inflammation-related gene expression

A well-described role for Orai1 in immune cells is the initiation of Ca$^{2+}$-dependent gene expression[19,20], but whether this function of Orai1 extends to astrocytes remains unknown. To address this question, we analyzed global gene expression changes via RNA sequencing. We stimulated WT or Orai1 cKO astrocytes with a low dose of TG (0.2 μM) to directly activate Orai1 channels, and PDBu to co-stimulate protein kinase C (PKC) to mimic the signaling arising from GPCRs such as P2Y and protease-activated receptors whose activation stimulates SOCE[23,29]. We reasoned that this approach by-passes the Ca$^{2+}$-independent signaling arising from receptor activation (e.g., involving the MAP kinases) that are also linked to many GPCRs and hence more narrowly examines Ca$^{2+}$-dependent Orai1 contributions to gene expression (Supplementary Fig. 1). Each experimental batch included both resting and stimulated samples from both genotypes and principal component analysis of the data showed that there were minimal batch or sex effects within our samples (Supplementary Fig. 4c). We found that other than *Orai1*, only three other genes of unknown function, *Rpl35a-ps2*, *Gm28438*, *Rpl35a-ps6*, were significantly altered between resting WT and Orai1 cKO astrocytes (Supplementary Fig. 4a). Thus, reduction in Orai1 expression does not substantively affect global gene expression in resting cells.

By contrast, after astrocytes were stimulated with TG+PDBu, large transcriptomic differences emerged between WT and Orai1 cKO cells as well as in WT cells between resting and stimulated conditions (Fig. 2a and Supplementary Fig. 4b). When compared directly, 1786 genes were expressed more in stimulated WT relative to Orai1 cKO cells, and 1949 genes were expressed less. The emergence of these differences only after CRAC channel activation indicates that while

Orai1 signaling does not substantially affect gene expression in quiescent astrocytes, it plays a large role in cellular responses following CRAC channel activation. Gene-set-enrichment (GSEA) and gene-ontology (GO) analysis of differentially expressed genes (DEGs) revealed that compared to the Orai1 KO stimulated group, WT stimulated cells were significantly enriched for Reactome and GO terms related to inflammation, immunity, and cytokine signaling (Fig. 2b, Supplementary Tables 1 and 2). In particular, we noted the strong bias in the GO terms related to IL-1, NFκB, and IFN-γ pathway genes (Fig. 2b–f and Supplementary Tables 1 and 2). Hierarchical clustering of all genes within the IL-1 signaling pathway (Fig. 2c, Supplementary Fig. 4h) revealed strong reductions in Orai1 cKO astrocytes in the expression of genes related to NFκB signaling, including *Nfkb*, *Rela*, *Chuk (IKKα)*, and *Ikbkb*, along with *Tollip, Il33, Smad3, Il1rap, Traf6*, and *Il1rl1* (Fig. 2c). Interestingly, comparison of DEGs to a published dataset[30] in which astrocyte reactivity was induced indirectly in vivo by intraperitoneal administration of the endotoxin lipopolysaccharide (LPS) showed substantial overlap between the two datasets (Supplementary Fig. 4f). This overlap suggests that Orai1 may be an important mediator of LPS inflammation-evoked gene expression in brain astrocytes.

To gain insight into how the above-described inflammatory pathway genes are diminished due to Orai1 deletion, we assessed transcription factor activation using the HOMER suite[31]. Not surprisingly, binding motifs for numerous calcium-dependent transcription factors were significantly enriched in the WT cells relative to cKO cells, particularly CREB and NFATc1. Analysis of genes regulated by these transcription factors using gene lists curated by the ENCODE project[32] showed that in stimulated WT cells, a cluster of CREB-dependent genes that were positively regulated by cell stimulation was diminished in

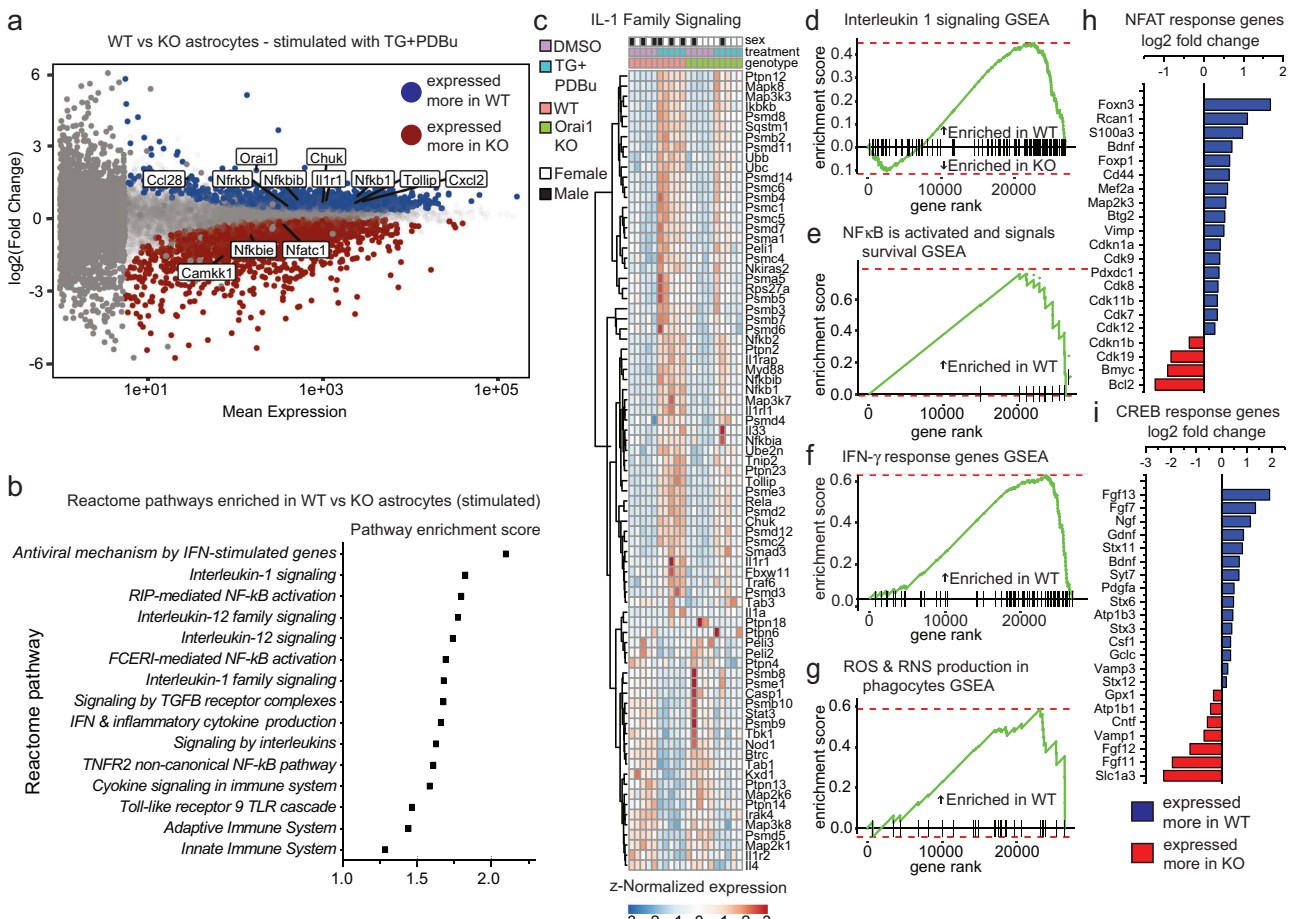

**Fig. 2 | RNA-sequencing reveals that stimulus-evoked inflammatory pathways are reduced in Orai1 cKO astrocytes.** Orai1$^{fl/fl}$ (WT) and Orai1$^{fl/fl\ GFAP-Cre}$ astrocytes were stimulated with a low dose (0.2 μM) of TG + 50 nM PDBu (or DMSO) for 6 h and differentially expressed genes were examined by RNA-seq analysis. **a** MA plot (log-intensity fold change vs log mean expression) showing differential gene expression between stimulated WT and Orai1 cKO astrocytes. Blue dots denote genes expressed significantly more in WT cells and red indicates genes expressed significantly more in Orai1 cKO astrocytes. Gray dots denote genes that were not significantly different between groups (p < 0.05, adjusted for multiple comparisons). **b** Pathway enrichment scores of selected reactome pathways related to inflammation. The x-axis indices show enrichment in WT-treated over KO-treated cells. **c** Heatmap of z-normalized within-sample expression of significantly variable genes in the "Interleukin-1 signaling" pathway from panel **b**. Both the treatment (TG +PdBu) and the genotype (WT vs Orai1 KO) have a strong influence on the relative expression of genes in this pathway. **d–g** Gene-set enrichment analysis (GSEA) in astrocytes following cell stimulation. The GSEAs indicate the extent to which all identified genes within the IL-1 signaling (**d**), NFκB activation (**e**), IFN-γ signaling (**f**), and ROS (**g**) reactome pathways are enriched in the WT or Orai1 cKO cells. Each vertical line on the x-axis represents a gene. **h, i** Enrichment of genes regulated by NFAT (**h**) or CREB (**i**) (shown on log$_2$ scale). NFAT- and CREB-response genes were collated from previous studies[33, 85–87]. n = 2 female, 3 male WT mice, 1 male, 4 female KO mice. Source data are provided as a Source Data file.

Orai1 KO cells (Supplementary Fig. 4g). NFAT response genes showed a similar pattern (Supplementary Fig. 4g). Astrocyte CREB activation is linked to transcriptional signatures related to metabolism and inflammation[33] and NFAT has been proposed to link calcium dysregulation and reactive astrogliosis[34]. Thus, these results suggest that Orai1 serves as an important upstream activator of Ca$^{2+}$-dependent transcription and plays a vital role in controlling the astrocyte inflammatory response. Interestingly, pathways related to mitosis and cell cycle checkpoints were also enhanced in WT-stimulated cells (Supplementary Tables 1 and 2). This trend, which also occurs in reactive astrocytes[35], is in line with evidence implicating Orai1 in cell proliferation[20,36]. Taken together, the results of the RNA-seq analysis show that stimulus-evoked Orai1 signaling in astrocytes promotes the expression of genes related to inflammation.

In addition to inflammatory and transcriptional regulators, pathways related to metabolism including "General metabolic regulation" and "ROS and RNS production" were also significantly different between WT and Orai1 cKO cells (Fig. 2g and Supplementary Tables 1 and 2). Because metabolic shifts are characteristic of the activation of

many types of immune cells[37,38], this finding suggested that Orai1 may regulate astrocyte metabolism.

## Orai1 signaling stimulates astrocyte metabolism

Ca$^{2+}$ signaling is known to stimulate cellular metabolism, particularly the conversion of energy stores into ATP through glycolysis and aerobic respiration[39]. As cell metabolism is closely linked to inflammation and cytokine production[40–42], we next addressed whether Orai1-mediated Ca$^{2+}$ signaling stimulates astrocyte metabolism. In line with this hypothesis, gene expression in several metabolic pathways was suppressed in stimulated Orai1 cKO astrocytes relative to stimulated WT astrocytes (Fig. 2g, 3a, Supplementary Tables 1 and 2). Notably, several genes vital to glycolysis were reduced, including hexokinase 2 *(Hk2)* (which catalyzes the rate-limiting first step of glycolysis, ADP-dependent glucokinase *(Adpgk)* which converts glucose to glucose-6-phosphate, and glucokinase regulatory protein *(Gckr)* which negatively regulates glucokinase enzymes (Fig. 3a). These alterations suggest that Orai1 Ca$^{2+}$ signaling is an important checkpoint for control of astrocyte metabolism.

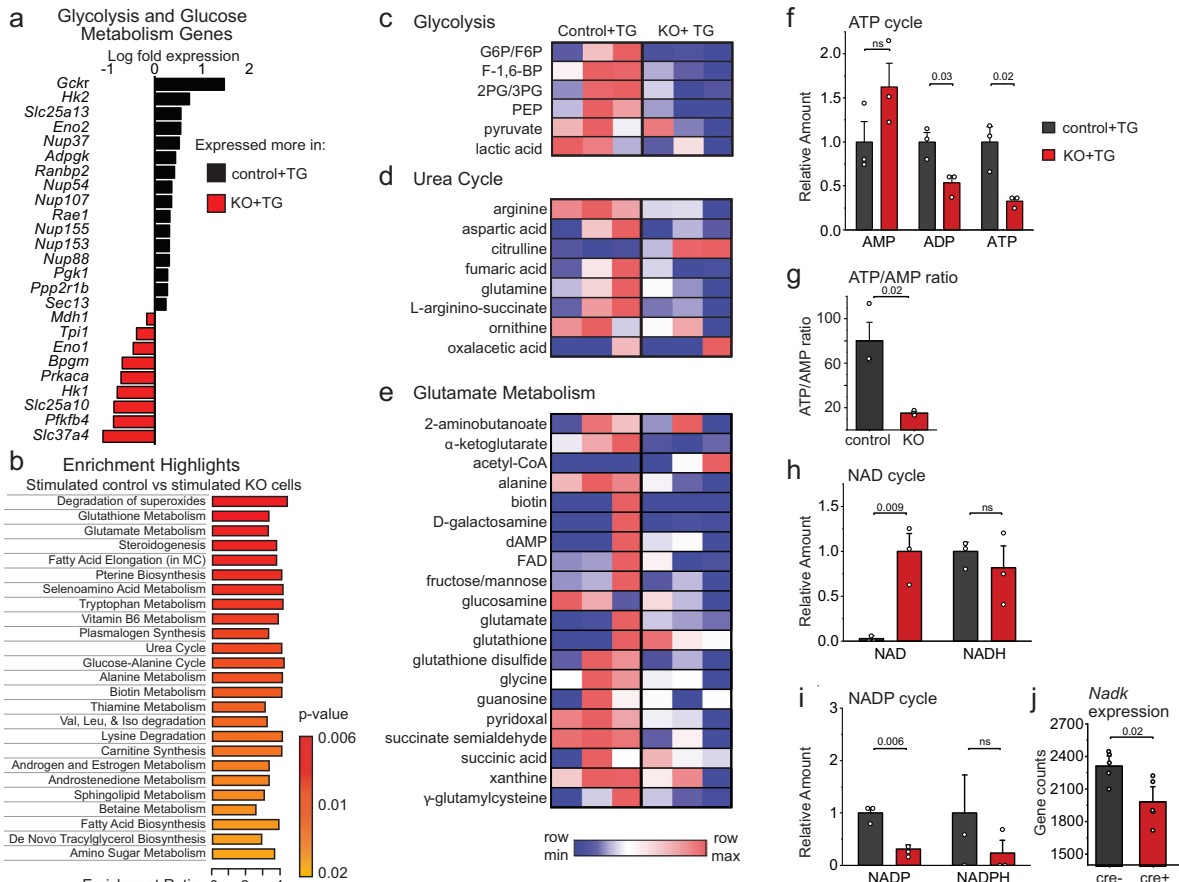

**Fig. 3 | Orai1 signaling stimulates metabolic reprogramming in astrocytes.**
**a** Differentially expressed genes (shown on $Log_2$ scale) related to glycolysis and glucose metabolism induced by Orai1 deletion. Astrocytes were stimulated with a low dose (0.2 μM) of TG + 50 nM PDBu for 6 h. **b** Enrichment highlights (of the 25 most enriched pathways) assessed by Quantitative Enrichment Analysis with the Global Test for statistical analysis (metaboanalyst.ca). Bars denote the enrichment ratio in stimulated control over Orai1 KO cells. **c–e** Relative metabolite quantities of individual metabolites in the glycolysis (**c**), glutamate metabolism (**d**), and urea cycles. Data are shown as a heat map with each box representing values from an individual astrocyte culture from one mouse. Red depicts the maximum and blue minimum for each row. $n = 3$ mice/group. **f** Relative levels of ATP, ADP, and AMP. Orai1 cKO astrocytes show reduced levels of ATP and ADP. **g** Within-sample ATP/AMP ratios. (**h, i**) Relative levels of $NAD^+$ and NADH, and $NADP^+$ and NADPH in astrocytes. **j** *Nadk* expression using gene counts from the RNAseq experiment in Fig. 2. ($n = 5$ mice/group). **b–i** $n = 3$ mice in each genotype. Bars indicate Mean +/- SEM. The sex of the pups was not determined. Statistical tests were conducted by two-tailed, unpaired T-tests. Source data are provided as a Source Data file.

To directly examine metabolic changes regulated by Orai1, we used mass spectrometry to define the metabolite profiles of control and Orai1 cKO astrocytes. Quantitative enrichment analysis (QEA) of metabolic pathways showed that in quiescent astrocytes, there was no difference between WT and Orai1 KO groups (Supplementary Fig. 5a). Furthermore, ATP and NAD(P) cycles were essentially unaffected (Supplementary Fig. 5d–g), indicating that in quiescent cells, Orai1 does not substantively regulate astrocyte metabolism.

Stimulation of Orai1 signaling in control astrocytes by TG +PDBu markedly enhanced multiple metabolic pathways, including pathways related to glutamate metabolism, the urea cycle, and the citric acid cycle (Supplementary Fig. 5b). Orai1 cKO astrocytes, however, showed downregulation of many metabolic pathways relative to control astrocytes, especially in pathways related to glycolysis, degradation of superoxides, glutamate metabolism, and the urea cycle (Fig. 3b and Supplementary Fig. 5c). Additionally, key glycolysis intermediates including fructose 1,6 bisphosphate, 2- and 3-phosphoglycerate, phosphoenolpyruvate, pyruvate, and lactate were reduced in Orai1 cKO astrocytes (Fig. 3c), in line with the results of the RNA-seq analysis showing that some glycolytic genes are reduced in Orai1 cKO astrocytes (Fig. 3a). Aspartic acid, fumaric acid, α-KG, and other metabolites involved in the urea, glutamate,

and TCA cycles also showed a downward trend in Orai1 cKO astrocytes, suggesting loss of Orai1 also destabilizes mitochondrial-based metabolic pathways (Fig. 3d, e).

We next examined ATP levels to determine how the metabolic needs of stimulated control and Orai1 KO astrocytes differ. ATP levels and the ATP/AMP ratio were significantly reduced in stimulated Orai1 KO astrocytes (Fig. 3f, g), which is in line with the finding that glycolysis pathways are reduced by Orai1 deletion (Fig. 3a, c). Even more strikingly, $NAD^+$ but not its reduced form, NADH, was dramatically elevated in Orai1 KO astrocytes (Fig. 3h), indicating that the $NAD^+$/NADH ratio is skewed towards an increased oxidative state in Orai1 KO cells. The phosphorylated counterparts, NADPH and $NADP^+$, were also reduced in Orai1 KO astrocytes (Fig. 3i), which could be due to diminished *Nadk* gene expression in the KO- vs WT-treated astrocytes (Fig. 3j) and which could indicate greater demands in WT cells for redox reactions due to increased metabolic activity. Taken together, these findings show that loss of Orai1 decreases the production of ATP, $NAD^+$ use, and key metabolic intermediates in the glycolysis, glutamate metabolism, and urea cycles. These results add to the growing evidence implicating SOCE in cell metabolism[43–45] and indicate that Orai1 plays a crucial role in the metabolic reprogramming that supports increased energy needs following astrocyte stimulation.

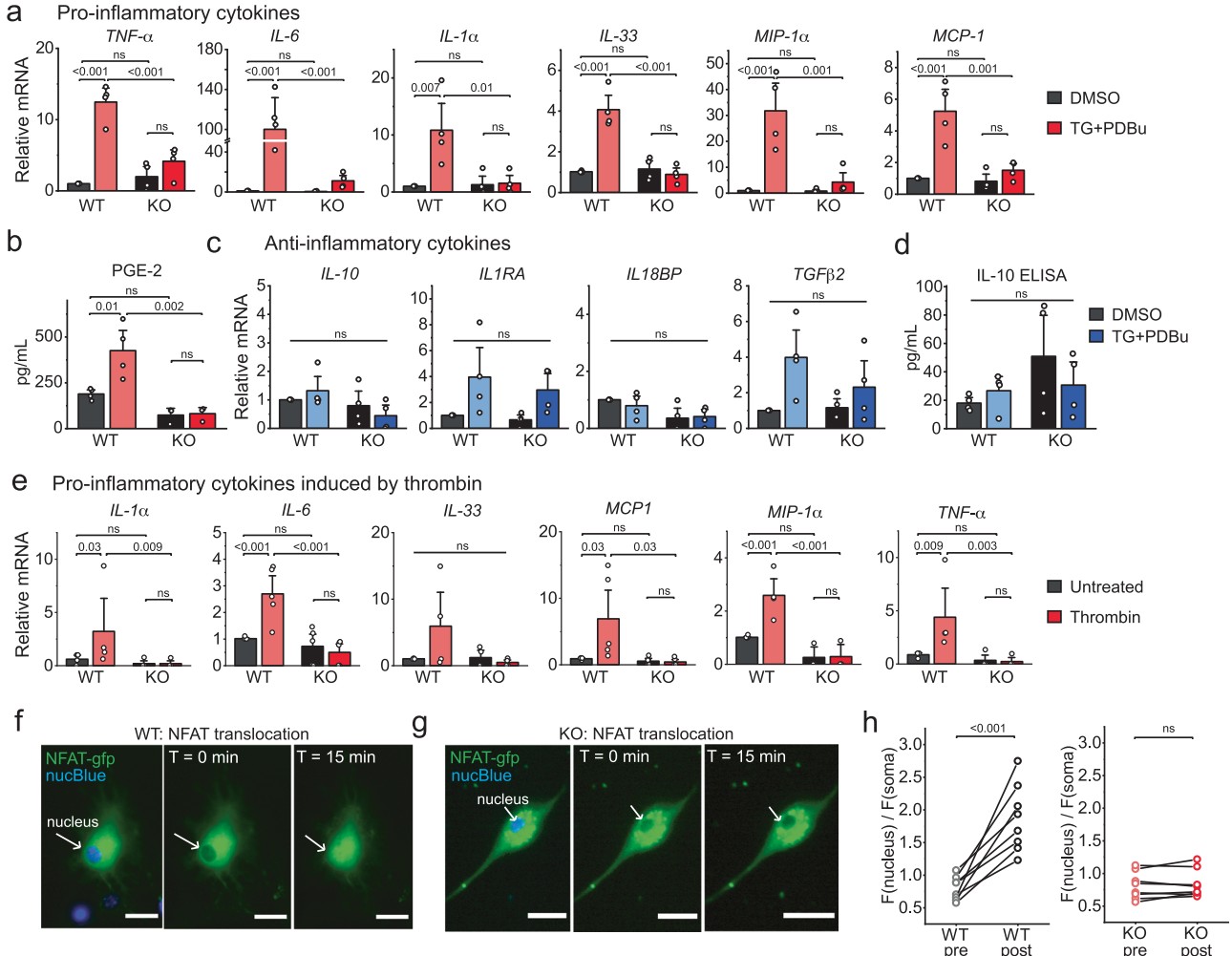

**Fig. 4 | Orai1 activation stimulates the production of pro-inflammatory cytokines from astrocytes. a** Induction of inflammatory cytokines and chemokines assessed by real-time PCR. WT (Orai1$^{fl/fl}$) and Orai1 KO (Orai1$^{fl/fl\ GFAP\text{-}cre}$) astrocytes were stimulated for 6 h with a low dose (0.2 μM) of TG + 50 nM PDBu and cytokines were measured via two-step real-time PCR. mRNA levels were normalized to GAPDH. ($n$ = 4 mice/group). **b** PGE-2 release from astrocytes measured by ELISA. ($n$ = 4 WT mice, 3 KO mice.) **c, d** Anti-inflammatory cytokines assessed by qPCR (**c**) and ELISA (**d**) ($n$ = 4 mice/group). **e** Induction of proinflammatory mediators by thrombin. Cytokines were measured by real-time PCR in astrocytes treated with thrombin (10 U/ml, 6 h) ($n$ = 5 WT mice, 4 KO mice). **f, g** Activation of CRAC channels induces nuclear translocation of NFAT-GFP. WT (Orai1$^{fl/fl}$) (**f**) or Orai1$^{fl/fl\ GFAP\text{-}cre}$ astrocytes (**g**)

astrocytes were transfected with NFATc3-GFP to monitor NFAT dynamics. Cells were incubated with NucBlue for 20 min prior to imaging to visualize the nucleus. Cells were stimulated for 15 min with TG (1 μM) in 2 mM Ca$^{2+}$ Ringer's solution to stimulate Orai1 activity. **h** Quantification of NFAT-GFP nuclear translocation (ratio of the mean GFP intensity in the nuclear and cytosolic regions). Data are displayed as the nuclear/cell GFP fluorescence ratio ($n$ = 8 cells from 3 mice/group). Scale bar = 10 μm. For all summary data, the bars indicate mean +/- SEM. For panel **a**: data separated by sex are provided in Supplementary Fig. 6. For panels **b**–**h**: the sex of the pups was not determined. Statistical tests were conducted by two-way ANOVA followed by Tukey posthoc tests for panels **a**–**e**, and paired t-test for panel **h**. Source data are provided as a Source Data file.

## Activation of Orai1 stimulates production of proinflammatory cytokines in astrocytes

The RNA-seq and metabolomics findings showing that ablation of Orai1 impairs inflammatory and metabolic signatures prompted us to directly examine whether Orai1 signaling induces proinflammatory cytokines. We addressed this question via realtime qPCR analysis of cytokine expression in primary hippocampal astrocytes from WT and Orai1 cKO astrocytes (Fig. 4a, c). This analysis revealed that activation of Orai1 channels stimulated the transcription of multiple inflammatory cytokines, including the classical mediators IL-1α, IL-33, IL-6, and TNFα as well as the chemokines, MCP1 and MIP-1α (Fig. 4a). Induction was either lost or strongly attenuated in Orai1 KO astrocytes from both male and female mice (Fig. 4a, Supplementary Fig. 6). These effects of Orai1 ablation on inflammatory cytokine production are similar to previously described effects of knocking out Orai1 in spinal microglia[24]. Likewise, the production of prostaglandin E2 (PGE$_2$) was stimulated by Orai1 in astrocytes (Fig. 4b) reminiscent of CRAC-

channel mediated induction of this major inflammatory mediator in epithelial cells[46]. By contrast, activation of Orai1 channels did not appreciably affect the production of anti-inflammatory cytokines (IL-10, IL13, IL-18BP, TGFβ2, and IL-1Ra) (Fig. 4c, d).

As noted earlier, a key proinflammatory agonist linked to Orai1 activation in astrocytes is the serine protease, thrombin[23]. Consistent with a role for Orai1 in the thrombin-mediated astrocyte inflammatory response, deletion or pharmacological suppression of Orai1 with BTP-2 impaired many cytokines induced by this agonist (Fig. 4e, Supplementary Fig. 7a). Moreover, as seen for direct CRAC channel activation by TG, deletion of Orai1 had no effect on anti-inflammatory cytokines (Supplementary Fig. 7b). Similarly, when stimulated by the endotoxin, lipopolysaccharide (LPS), hippocampal astrocytes showed strong increases in release of inflammatory cytokines including IL-1α and IL-6 and these increases in cytokine release were reduced by inhibition of Orai1 (Supplementary Fig. 7a). Deletion of Orai1 did not affect the expression of numerous molecules in the LPS and inflammasome

signaling cascade including mRNA levels of TLR4, P2X receptors, pannexins, connexins, and related molecules that are tied to ATP signaling and its release (Supplementary Fig. 8). Thus, these results indicate that Orai1 functions as an important checkpoint for transcriptional production of proinflammatory cytokines in response to a wide range of agonists. The in vivo relevance of these effects for the intact brain is examined in later sections below.

Synthesis of proinflammatory cytokines including IL-6, TNFα, and IL-2 occurs in large part through activation of the transcription factors, NFAT and NF-κB[46,47]. We observed that nuclear translocation of GFP-NFATc3, which we employed an optical readout for NFAT activation, was abolished in Orai1 cKO astrocytes (Fig. 4f–h). Likewise, TG- and thrombin-induced NFAT-mediated gene expression assessed by NFAT-luciferase reporter activity was largely abolished in the Orai1 KO astrocytes (Supplementary Fig. 7c,e). These results are consistent with the RNAseq analysis (Fig. 2) indicating that NFAT-dependent gene expression signatures are impaired in Orai1 cKO cells. Additionally, an NFκB-luciferase reporter assay showed that NFκB-dependent gene expression evoked by SOCE was strongly impaired in Orai1 KO astrocytes (Supplementary Fig. 7d, f). Inhibiting NFAT activation with cyclosporin A (CsA) blocked the induced expression of IL-1α, MIP-1α and MCP1 (Supplementary Fig. 7g). By contrast, the NFκB inhibitor, BMS[48], did not affect induction of IL-1α and MCP1 but strongly inhibited the production of MIP-1α and modestly inhibited IL-6 (Supplementary Fig. 7g). Taken together, the results of these mechanistic studies indicate that Orai1-mediated SOCE stimulates NFAT- and NFκB-dependent gene expression to drive proinflammatory cytokine synthesis in hippocampal astrocytes.

## Astrocyte Orai1 signaling promotes neuroinflammation in vivo

Our results thus far indicate that Orai1-mediated Ca²⁺ signaling drives a pro-inflammatory cytokine response in astrocytes. What are the implications of this response for CNS inflammation in vivo? We addressed this question using a widely employed model of neuroinflammation in which LPS, a component of the outer membrane of gram-negative bacteria and a potent activator of TLR4 receptors, activates the peripheral immune system through stimulation of macrophages, dendritic cells, neutrophils, and mast cells. In both mice and humans, the robust peripheral inflammation caused by the administration of LPS induces sickness behaviors in the hours following LPS injection, followed by subsequent symptomatology of depression-like behaviors that can last for several days[6,49,50]. Although LPS stimulates astrocyte cytokine production in vitro in an Orai1-dependent manner (Supplementary Fig. 7a), it is important to note that LPS does not appreciably cross the blood-brain barrier (BBB)[51]. Rather, the brain effects and behavioral changes of peripheral LPS administration are thought to be induced by an influx of cytokines and infiltration of activated immune cells crossing into the brain to drive neuroinflammation and depression-like behaviors[52–55]. We examined the potential contributions of Orai1-dependent astrocyte reactivity for neuroinflammation to this systemic inflammatory stimulus.

As a first step, we compared the expression of glial reactive markers GFAP and IBA1 in vivo in WT and cKO mice in the hippocampus 48-hours after LPS injection. As before, we focused these studies on the hippocampus as the hippocampus is highly susceptible to inflammatory disease[50] and peripheral LPS exposure is well-known to cause upregulation of a range of inflammatory markers in the hippocampus[56]. In naïve Orai1 cKO mice, GFAP and IBA1 expression assessed via confocal microscopy of immunostained brain sections were not appreciably different from naïve WT mice, indicating that the loss of Orai1 does not directly affect baseline inflammation in the hippocampus (Fig. 5, Supplementary Fig. 9). However, LPS administration led to increases in GFAP and IBA1 expression in WT mice in several areas of the hippocampus (CA1, CA3, DG) and this occurred in both sexes (Fig. 5b–d & Supplementary Fig. 9). By comparison, LPS-induced increases of GFAP and IBA1 were lower in Orai1 cKO mice of both sexes (Fig. 5b–d & Supplementary Fig. 9). Thus, conditional deletion of Orai1 in astrocytes blunts peripheral inflammation-induced astrocyte and microglial reactivity.

GFAP and IBA1 are generalized markers for activated astrocytes and microglia and cannot distinguish neuroprotective from neuroinflammatory responses. Hence, we also assessed levels of C3, a member of the complement cascade whose increased expression and release from astrocytes directly correlates with neuroinflammation[3, 57]. Immunohistochemistry revealed that C3 expression is upregulated in WT LPS-treated mice compared to saline-treated controls (Supplementary Fig. 10a, b). By contrast, C3 levels in Orai1 cKO mice were comparable to those seen in unchallenged WT mice (Supplementary Fig. 10b). Because C3 produced by astrocytes activates neighboring microglia[58], a reduction in C3 may explain why microglial markers (IBA1) are diminished in astrocyte Orai1 KO mice and supports the emerging viewpoint that astrocytes operate in close coordination with microglia to amplify cascades of inflammation[59]. Importantly, in homogenized hippocampal brain tissue from WT mice, we found that LPS administration also increased levels of IL-6 and IL-1α, two cytokines that are well established drivers of neuroinflammation in vivo[60,61] (Fig. 5e), and these increases were blunted in Orai1 cKO (Orai1^fl/fl Aldh1l1-Cre/ERT2+tamoxifen) mice (Fig. 5e). Taken together, these results argue that astrocytes play an important role in amplifying levels of inflammatory cytokines in the brain following LPS challenge and this regulation is controlled by Orai1 calcium channels.

As noted above, brain inflammation following peripheral LPS administration is thought to be triggered by inflammatory factors crossing into the brain through the BBB[51]. Among these mediators, thrombin has well-defined roles in inducing brain inflammation through its potent effects in stimulating microglia and astrocytes[25]. Because thrombin activates Orai1 channels in astrocytes to stimulate the production of inflammatory mediators from astrocytes (Supplementary Fig. 2d–f, Fig. 4e, Supplementary Fig. 7), we next assessed the effects of LPS on levels of brain thrombin. 48 h after LPS administration, hippocampal brain lysates showed strong elevations in thrombin levels relative to controls (Fig. 5f). Moreover, the increase in thrombin levels was similar between WT and astrocyte Orai1 cKO mice, indicating that deletion of Orai1 in astrocytes has no impact on LPS-induced thrombin increases in the brain (Fig. 5f). SDF1α, a chemokine that stimulates SOCE[36], also trended higher in LPS-treated mice (Fig. 5f). These results indicate that LPS enhances in vivo brain levels of the SOCE activator, thrombin, and possibly other mediators which could stimulate astrocyte Ca²⁺ signaling to enhance brain inflammation.

## Ablation of Orai1 decreases LPS-induced astrocyte Ca²⁺ signaling in situ

Our previous work has shown that, in their native environment in the brain, the intricate branches and fine processes of astrocytes display spontaneous and GPCR-evoked Ca²⁺ fluctuations which are regulated by Orai1 calcium channels[23]. To examine how these Ca²⁺ signals are affected by peripheral LPS-induced inflammation, we used the genetically encoded Ca²⁺ indicator GCaMP6f to image Ca²⁺ fluctuations using 2-photon laser scanning microscopy (2PLSM) in acutely prepared brain slices (Fig. 6a). GCaMP6f was selectively expressed in hippocampal CA1 astrocytes by injecting the dorsolateral hippocampus in adult mice (P50-65) with an AAV5 viral expression vector carrying the GfaABC₁D astrocyte-specific promoter (Fig. 6a, b)[62]. Following a two-three week recombination period, we euthanized mice and studied astrocyte Ca²⁺ signals in CA1 stratum radiatum layer as using 2PLSM as previously described[23].

GCaMP6f-expressing astrocytes imaged under 2PLSM displayed an intricate pattern of astrocyte arborization expected of astrocytes in their native environment in the hippocampus (Fig. 6b). We mapped Ca²⁺ fluctuations in three anatomically defined compartments: the

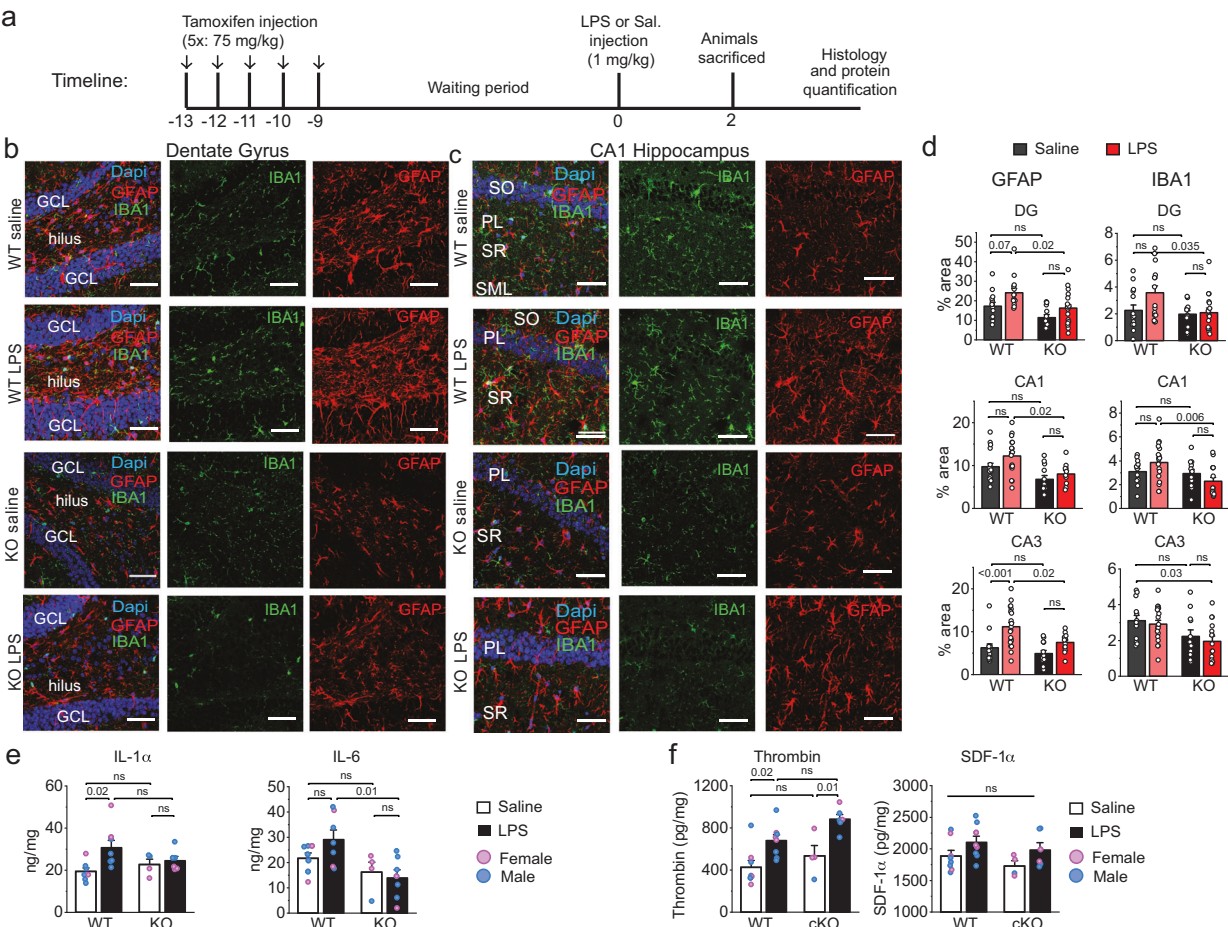

**Fig. 5 | Expression of neuroinflammatory markers following LPS challenge is reduced in Orai1 cKO mice. a** A schematic of the protocol used for the experiment. Orai1$^{fl/fl}$ (WT) and Orai1$^{fl/fl Aldh1l1-Cre/ERT2}$ mice were injected with tamoxifen for five days to induce the deletion of Orai1. LPS or saline was delivered as indicated on day 10 following the last tamoxifen injection. Animals were sacrificed 2 days later and analyzed by immunohistochemistry and ELISA. **b, c** Immunohistochemistry of hippocampal GFAP (to label astrocytes) and IBA1 (to label microglia) in the dentate gyrus and the CA1 regions of the hippocampus. The images shown were obtained from male mice; female mice showed a similar pattern (Supplementary Fig. 8). Scale bar = 50 µm. **d** Quantification of GFAP and IBA1 expression assessed by measuring the % area of the region of interest occupied by fluorescent signal. Data are given as means +/- SEM for $n$ = 14–18 images from 3 WT saline mice, $n$ = 14–20 images from 5 WT LPS mice, $n$ = 11–13 images from 3 KO saline mice, $n$ = 10–20 images from 4 KO LPS mice. **e** Levels of IL-1α and IL-6 measured in homogenized hippocampal tissue

lysates by ELISA. (IL-1α - WT saline group: $n$ = 5 male, 2 female mice; WT LPS group: $n$ = 6 male, 2 female mice; KO saline group: $n$ = 2 male, 2 female mice; KO + LPS group: $n$ = 3 male, 3 female mice. IL-6 - WT saline group: $n$ = 5 male, 2 female mice; WT LPS group: $n$ = 5 male, 2 female mice; KO saline group: $n$ = 1 male, 3 female mice; KO LPS group: $n$ = 5 male, 2 female mice. **f** Peripheral LPS administration increases thrombin levels in the brain. Thrombin (A) and SDF-1α (B) were measured via ELISA in homogenized hippocampal tissue lysates in saline (white bars) and LPS-injected (black bars) mice 48 h after injection. Thrombin is significantly elevated in LPS-treated WT and cKO mice, while SDF1α is modestly elevated. (WT saline group: $n$ = 3 female and 5 male mice. WT LPS group, $n$ = 3 female and 5 male mice. Orai1 cKO saline group: $n$ = 3 female, 1 male mice, Orai1 cKO LPS group: $n$ = 4 male, 2 female KO mice). All data are given as mean +/- SEM. Statistical tests were conducted for the pooled data by two-way ANOVA followed by Tukey posthoc tests in each graph. Source data are provided as a Source Data file.

soma, the primary proximal processes coming off the cell body, and the distal tertiary processes (Fig. 6b, Supplementary Fig. 11a, b). Both the soma and the astrocytic processes showed on-going spontaneous activity seen as transient rises in Ca$^{2+}$ which occurred at average frequencies of ~0.6–2 events/min in saline exposed mice (Fig. 6c, d, Supplementary Movie 1). Importantly, ablation of Orai1 did not affect the frequency or amplitude of the baseline Ca$^{2+}$ fluctuations in saline-injected mice, indicating that the spontaneous Ca$^{2+}$ oscillations are Orai1-independent in healthy mice (Fig. 6d, Supplementary Movie 2). By contrast, as previously shown[23], upregulation of Ca$^{2+}$ fluctuations seen in WT astrocytes by thrombin was lost in Orai1 cKO brain slices (Supplementary Fig. 11).

To examine how these spontaneous Ca$^{2+}$ oscillations are altered in response to peripheral LPS-mediated inflammation, we examined astrocyte Ca$^{2+}$ signaling using 2PLSM 18–24 h after LPS administration. LPS-treated WT mice showed highly significant increases in the frequency of Ca$^{2+}$ fluctuations in the soma as well as the proximal and

distal processes (Fig. 6c, d and Supplementary Movie 3). In the soma, the Ca$^{2+}$ fluctuation frequency increased from 0.83 ± 0.09 in saline-treated controls to 2.18 ± 0.22 events/min in LPS-treated mice ($p$ = 2.5 × 10$^{-7}$). Likewise, robust increases in the frequency of spontaneous Ca$^{2+}$ fluctuations were readily detected in the proximal and distal astrocyte processes. There was no change in the amplitude of the Ca$^{2+}$ events (Fig. 6e). Thus, LPS-mediated peripheral inflammation causes a robust increase in astrocyte Ca$^{2+}$ signaling frequency in situ in the hippocampus. In marked contrast to these responses in WT mice, Orai1 cKO mice failed to show increases in the frequency of Ca$^{2+}$ fluctuations following LPS challenge (Fig. 6c, d and Supplementary Movie 4). Instead, in these mice, the astrocyte Ca$^{2+}$ oscillation frequency following LPS challenge remained similar to frequencies in saline-injected controls (Fig. 6c, d). Thus, these results indicate that peripheral LPS administration increases Orai1-mediated astrocyte Ca$^{2+}$ signaling in the hippocampus, likely due to stimulation of astrocytes by thrombin and other mediators that are induced by LPS, providing

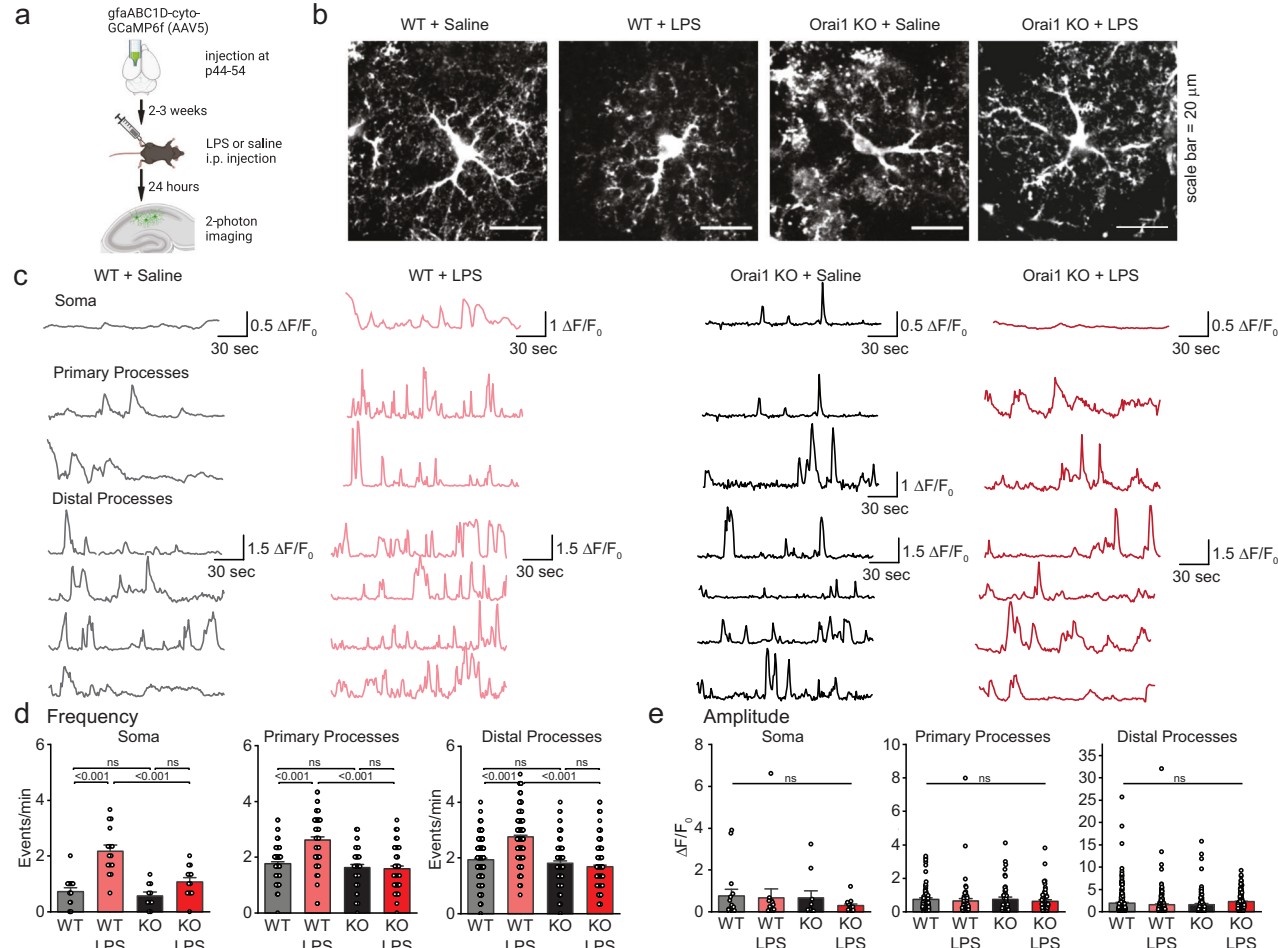

**Fig. 6 | LPS-evoked increases in astrocyte Ca²⁺ signaling are abolished in Orai1 cKO mice. a** Schematic illustrating experimental protocol. GCaMP6f was expressed in astrocytes of the hippocampus through stereotaxic injections of an AAV5 virus with an astrocyte-specific gfaABC1D promoter. After allowing 2-3 weeks for GCamp6f expression, mice were intraperitoneally injected with either 1 mg/mL LPS or equivalent volume of saline. 24-hours following intraperitoneal LPS administration, Ca²⁺ fluctuations in astrocytes expressing GCaMP6 were imaged using 2PLSM in the *Stratum Radiatum* region of the CA1 hippocampus. **b** 2-P images of astrocytes transfected with gCAMP6f. Each image is the maximum intensity projection of the time series (180 s). Scale bar = 20 μm. **c** Traces of the *ΔF/F_O* values from the soma, primary branches, and distal branches of astrocytes in brain slices from WT and Orai1 cKO mice. Mice were administered either saline or LPS as

indicated. Sample images and traces are representative of experiments performed on 10–16 cells/group from 3 mice/group. **d, e** Summary graphs of the frequency (**d**) and amplitude (**e**) of calcium transients calculated over three minutes of imaging. Data are presented as mean values +/- SEM. Each dot represents one ROI (amplitude is the average of all *ΔF/F_O* peaks in one ROI, frequency is the total # of events/3 min), and statistical tests were performed on the mean amplitude/frequency of all ROIs pertaining to a given cell. WT saline: $n = 16$ cells from 2 male, 1 female mice; WT + LPS: $n = 15$ cells from 2 female, 1 male mice; cKO+saline: $n = 10$ cells from 1 female, 2 male mice; cKO+LPS: $n = 13$ cells from 1 female, 2 male mice. Statistical tests were conducted by two-way ANOVA followed by Tukey posthoc tests for each graph. Illustrations created with BioRender.com. Source data are provided as a Source Data file.

an important mechanistic link between systemic inflammation and astrocyte activation in the CNS.

**Astrocyte Orai1 channels regulate inflammation-evoked changes in neuronal activity**

The results presented thus far indicate that systemic inflammation by LPS increases astrocyte Ca²⁺ fluctuations and inflammatory cytokines in the brain in an Orai1-dependent manner. What are the implications of this Orai1-regulation of astrocyte Ca²⁺ signaling and brain inflammation for neural activity? We addressed this question using patch-clamp electrophysiology to analyze the basic features of synaptic transmission in hippocampal brain slices by recording spontaneous excitatory and inhibitory currents (sEPSCs and sIPSCs) from CA1 pyramidal neurons. Mice were administered a single dose of LPS intraperitoneally and sEPSC and sIPSC activity was recorded 18–20 h after LPS injections (Fig. 7a). As slice perfusion likely washes out soluble factors present in the intact brain, we also examined the effects of adding back thrombin to the brain slices. We have previously shown

that thrombin activation of astrocytes enhances GABAergic interneuron activity in the hippocampus in a manner that requires astrocyte Orai1 channels[23].

These electrophysiological measurements revealed several interesting LPS- and genotype-evoked differences between WT and Orai1 astrocyte cKO mice. First, examination of inhibitory neurotransmission revealed that LPS-treated Orai1 cKO mice showed sharp decreases in both the amplitude and frequency of sIPSCs, which did not occur in WT mice (Fig. 7e, g). Cumulative histogram plots of the responses indicated substantial lengthening of the inter-event intervals in Orai1 cKO mice but not in WT mice, consistent with the overall decrease in the frequency of sIPSCs (Fig. 7f, h). Thus, loss of astrocyte Orai1 channels results in strong decreases in inhibitory neurotransmission in the CA1 hippocampus following LPS administration. As described previously[23], thrombin-evoked increases in the frequency of sIPSCs in naïve WT mice were lost in Orai1 cKO slices with no other changes in other features of synaptic transmission (Supplementary Fig. 12a–d), suggesting that loss of astrocyte-derived neuroactive mediators may

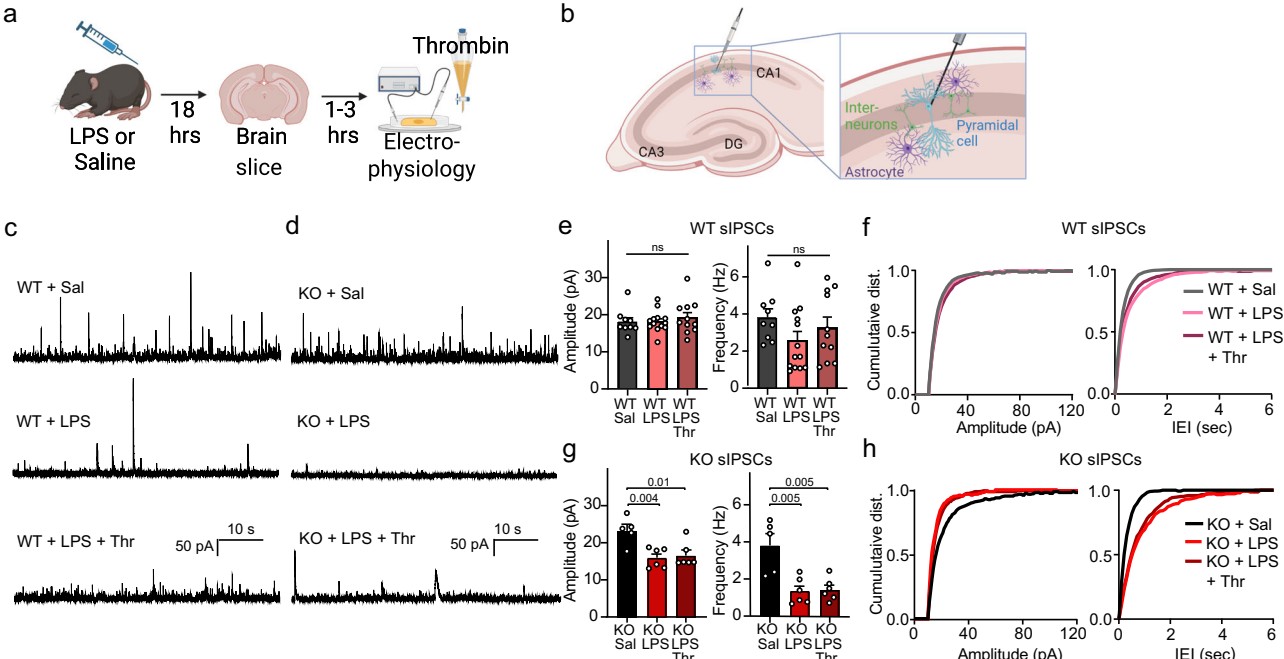

**Fig. 7 | Astrocyte Orai1 cKO mice show distinct changes in inhibitory synaptic transmission following LPS challenge. a** Schematic of the experimental timeline. WT and Orai1 cKO mice were injected with LPS intraperitoneally and brains were removed for slice electrophysiology -18-20 h later. **b** Schematic of the slice-patch clamp recordings carried out in panels *C-H*. Whole-cell recordings were carried out in CA1 pyramidal neurons to record excitatory and inhibitory postsynaptic currents. **c**, **d** Example traces of sIPSCs in the indicated conditions and genotypes. **e**–**h** Summary graphs of the amplitude and frequency of sIPSCs. The amplitude and frequency were reduced in LPS administered Orai1 cKO mice but not WT mice. Data are presented as mean values +/- SEM. (**e**, **g**). Cumulative distribution graphs showing amplitude and inter-event interval (IEI) for WT (**f**) and cKO (**h**) mice illustrate the effect of LPS on Orai1 cKO mice. The number of recordings were as follows: n = 9 WT cells from 3 male & 1 female mouse, n = 13 WT + LPS cells from 2 female & 3 male mice, n = 11 WT + LPS+Thr cells from 2 female & 3 male mice, n = 5 cKO cells from 4 male mice, n = 6 cKO+LPS cells from 2 male & 1 female mice, n = 6 cKO+LPS+Thr cells from 2 male & 1 female mice. Statistical tests were conducted by one-way ANOVA followed by Tukey posthoc tests for all comparisons. Illustrations created with BioRender.com. Source data are provided as a Source data file.

partly explain the decline in sIPSCs in the cKO mice. Second, examination of excitatory neurotransmission showed that LPS treatment modestly increased the amplitude of sEPSCs in WT mice, especially in the presence of thrombin, and this enhancement did not occur in Orai1 cKO mice (Supplementary Fig. 12f–i). Orai1 KO slices also showed a sharp drop in sEPSC frequency following LPS administration which was reversed by thrombin (Supplementary Fig. 12f–i). Inflammatory cytokines such as TNFα are known to enhance excitatory synaptic transmission[63], which may explain the potentiation of sEPSCs and reversal in Orai1 cKO mice. These electrophysiological results indicate that loss of Orai1 astrocyte $Ca^{2+}$ signaling diminishes inhibitory tone in the CA1 hippocampus after LPS administration. Previous studies have shown that some types of rapidly acting antidepressants evoke similarly strong decreases in inhibitory neurotransmission in many regions of the brain including the hippocampus[64], raising the possibility that ablation of Orai1 may also evoke antidepressant effects.

**Deletion of Orai1 in astrocytes mitigates inflammation-induced depression-like behaviors**
High levels of circulating cytokines are linked to a wide range of negative psychological changes in humans (and mice) including decreased activity, depression, and loss of energy[7, 10]. Given the above findings indicating that peripheral inflammation induces robust neuroinflammation in the brain that is mitigated in the Orai1 cKO mice, we next evaluated whether behavioral measures of inflammation-evoked depression are affected in the LPS-treated Orai1 cKO mice. Previous studies have shown that the global inflammation in rodents induced by LPS elicits general sickness and peripheral inflammation within hours of injection, which peaks around 6 h and subsides by 24 h post-injection[7], and is followed by changes in motivational behaviors that mirror depression in humans,

including anhedonia (inability to feel pleasure) and helplessness[49,65]. In mice, anhedonia is commonly assessed using a sucrose preference test (SPT), a reward-based test that exploits the innate interest of mice for sweet foods[66]. A reduced preference for sugar water (relative to water) indicates anhedonia. Helplessness is commonly assessed using the forced swim test (FST) and tail suspension test (TST) that measures escape-related mobility of mice as a metric for behavioral despair[66]. Importantly, all three tests are widely used for screening the efficacy of clinically used antidepressants in the LPS model of depression[65]. Therefore, we used these tests to address the role of Orai1 channels in mediating LPS-evoked depression behaviors in mice. We also examined WT and cKO mice for defects in general locomotion (open-field), working memory (Y-maze), general anxiety (zero-maze), and learning and memory (fear conditioning) (Fig. 8, Supplementary Fig. 13).

Six hours after LPS administration, both WT and cKO mice showed signs of sickness as measured by murine sepsis scores (MSS) (Fig. 8f) accompanied by reduced locomotion in the open field test (Fig. 8g). This was observed in both male and female mice but subsided ~24 h post-injection (Fig. 8f, g and Supplementary Fig. 13f, g) indicating that acute measures of sickness subside shortly after LPS injection and are similar in WT and Orai1 cKO mice. Male WT mice administered with LPS showed a significantly decreased preference for sucrose-water compared to saline-treated controls (Fig. 8c). Moreover, male WT mice treated with LPS showed increased immobility in the FST and TST when compared to saline-injected mice both at 24 h and 48 h following LPS administration (Fig. 8d, e, Supplementary Fig. 14b–d). These results align with previous studies indicating that LPS induces depression-like behaviors in male mice characterized by anhedonia and helplessness[49]. In marked contrast to these trends, however, Orai1 cKO mice maintained their preference for sweetened water in the

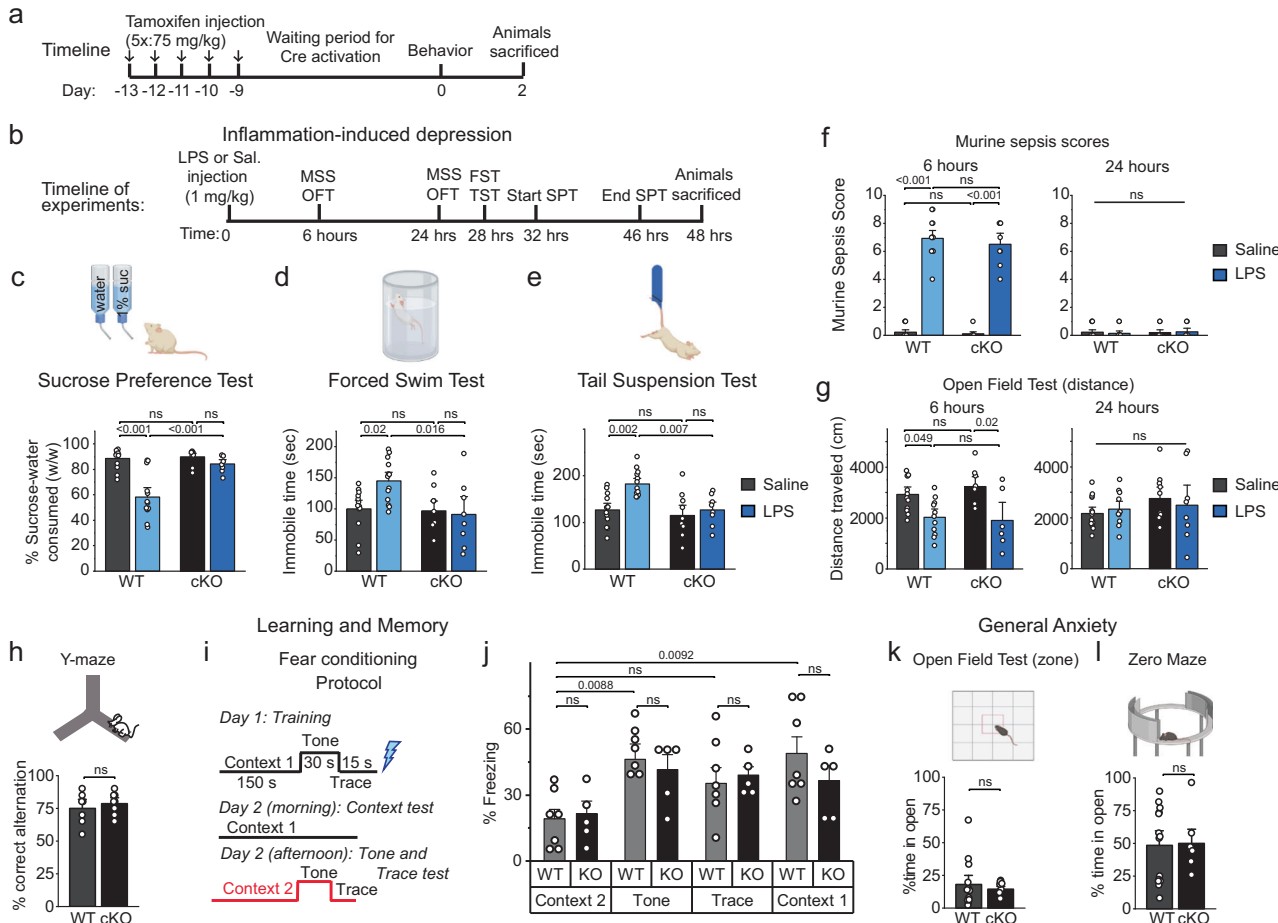

**Fig. 8 | Astrocyte-specific Orai1 KO mice are protected in a model of inflammation-induced depression-like behaviors. a, b** Schematics illustrating the timeline of the experiments. Tamoxifen was injected intraperitonially (i/p) for Cre-induced deletion of Orai1. LPS or saline was administered i/p nine days later to induce systemic inflammation. Tests for general locomotion (open-field) and depression-like behaviors (sucrose preference test (SPT), forced swim test (FST), and tail suspension test (TST)) were performed beginning 24 h after the LPS administration. **c**–**e** Analysis of depression-like behaviors following LPS administration. WT male mice consistently showed depression-like behaviors after administration of LPS, manifested as significantly decreased sucrose preference scores in the SPT and increased immobility times in the FST and TST. Orai1 cKO mice did not show these changes ($n = 13$ WT Sal and 13 WT LPS mice, 10 KO Sal mice, $n = 8$ KO LPS mice). **f, g** Mouse sickness behavior analysis including murine sepsis scores (MSS) (**f**) and general locomotion (**g**). MSS is increased and general

locomotion is decreased six hours post LPS administration in both WT and cKO mice but normalizes at 24 h. ($n = 14$ WT saline, $n = 13$ WT LPS, $n = 10$ Orai1 cKO saline, $n = 8$ Orai1 cKO LPS.) **h** Astrocyte Orai1 cKO mice do not show deficits in working memory as assessed in the Y-maze ($n = 7$ WT, $n = 8$ cKO). **i, j** Astrocyte Orai1 cKO mice do not show deficits in associative memory as assessed in the fear conditioning test. Fear conditioning was quantified by measuring the freezing duration in the indicated context. ($n = 7$ WT, $n = 5$ cKO) **k, l** cKO mice also show no deficits in general anxiety as measured by time spent in exposed areas of the Open Field (**k**) and Zero Maze (**l**). ($n = 14$ WT, $n = 10$ cKO *(OFT)*, $n = 8$ cKO *(Zero Maze)*). All data are presented as mean values +/- SEM. All data are from male mice. Statistical tests were conducted by two-way ANOVA followed by Tukey posthoc tests in each graph. Illustrations created with BioRender.com. Source data are provided as a Source Data file.

sucrose water test (Fig. 8c). Likewise, Orai1 cKO mice did not show increased immobility times in the forced swim and tail-suspension tests following LPS administration, in contrast to the increase in helplessness behavior in WT mice (Fig. 8d, e). Thus, conditional deletion of Orai1 in astrocytes protects male mice against inflammation-induced anhedonia, helplessness, and despair.

Do these behavioral changes in the astrocyte Orai1 KO mice reflect generalized alterations in learning and or motor ability? In the open-field test, neither LPS exposure nor genotype led to gross motor deficits compared to the control group as evidenced by distance traveled in the field at the 24-hr time point (Fig. 8g, right panel), indicating that LPS does not diminish general locomotion and exploratory behavior after 24 h, and further, that deletion of Orai1 in astrocytes does not affect gross motor mobility or the interest of mice in exploring their environment. Moreover, in contrast to the deletion of Orai1 in neurons which impairs learning and memory[67], conditional deletion of Orai1 in astrocytes did not affect working

memory performance in the Y-maze (Fig. 8h) or associative memory in the fear conditioning test (Fig. 8i, j). Additionally, in the open-field test neither LPS exposure (24-hours later) nor genotype (Orai1 cKO vs WT) elicited differences in gross motor or exploration, both in terms of the distance traveled in the field (Fig. 8g) or time spent in the middle part of the maze (Fig. 8k). Likewise, no genotype differences could be detected in the zero-maze test (Fig. 8l). Thus, deletion of astrocyte Orai1 confers relatively specific protection against behavioral depression without causing global impairments in locomotion, anxiety, exploratory behavior, or cognition, and moreover, reaffirm that the LPS-evoked depression is not simply a reflection of sickness behavior. Taken together, these results show that Orai1 channels in astrocytes are important regulators of astrocyte-mediated depression-like behaviors.

Female mice, in contrast to male mice, did not exhibit depression-like behaviors induced by LPS in any of the three tests we examined (Supplementary Fig. 13c-e) although female mice showed

sickness behaviors that were no different from male mice (Supplementary Fig. 13f, g). Since no behavioral phenotype was seen in female WT mice, the comparison between the WT and Orai1 KO groups was not instructive (Supplementary Fig. 13c-e). Literature on the expression of LPS-induced depression behaviors in female mice is limited as most previous studies did not document effects in female mice. However, several studies have found that female rodents do not respond to LPS-induced peripheral inflammation in the same manner as male mice, and specifically, do not show depression-like behaviors after LPS administration[50, 68–70]. Our own data indicate that while WT female mice exposed to LPS do show elevated levels of the neuroinflammatory markers, GFAP and IBA1, deletion of Orai1 dampened this upregulation (Supplementary Fig. 9), arguing that the lack of behavioral phenotype is not due to the absence of an inflammatory response in female mice to LPS but due to other protective factors. Recent work has shown that brain-derived nerve factor (BDNF) protects female mice from depression-like behaviors after LPS exposure[71], which may explain the lack of behavioral effect seen here in Orai1 cKO female mice.

## Discussion

In the present work, we have pursued an in-depth examination of the contributions of Orai1 channels for astrocyte Ca²⁺ signaling, gene expression, metabolism, in vivo brain inflammation, and inflammation-mediated depression-like behaviors. Our results show that ablation of Orai1 in astrocytes suppresses the ability of astrocytes to produce and release proinflammatory cytokines, dampens astrocyte metabolism, suppresses inflammation-driven astrocyte Ca²⁺ signaling, mitigates neuroinflammation in vivo in the hippocampus, and blocks the development of depression-like behaviors following intense peripheral inflammation all without substantively affecting baseline astrocyte functions or behavioral endpoints in naive mice (summarized in Supplementary Fig. 15). Taken together, these results show that astrocyte Orai1 signaling regulates multiple interlinked cellular processes that impact neuroinflammation in mice, identifying astrocyte Orai1 as potential target for modulation of inflammation-evoked brain inflammation and depression.

In response to stress or brain injury, astrocytes undergo a process of functional and structural remodeling that elicits a reactive phenotype whose molecular basis is not well understood[2,4]. The results presented in this study suggest that Orai1 plays an important role in remodeling astrocytes into the reactive state. Deletion of Orai1 impaired astrocyte gene expression in response to depletion of ER Ca²⁺ stores, especially in inflammatory pathways mediated by interferons, and upstream transcriptional inducers including NF-κB, and NFAT (Supplementary Fig. 15). Many cytokines found to be dampened in Orai1 cKO astrocytes including IL-1α, IL-33, TNFα, IL-6, and MCP1 are implicated in mediating neuroinflammation in disease states such as multiple sclerosis, Alzheimer's disease, and stroke[53,55] suggesting that Orai1-driven astrocyte activation may have a role in exacerbating neurological disease. Moreover, we observed that downregulation of inflammatory pathways in Orai1 cKO astrocytes is accompanied by alterations in cellular metabolism, including decreased ATP production, an increase in AMP/ATP ratio, and a shift towards an oxidized NAD⁺/NADH ratio. Because increased metabolism and reduced NAD⁺/NADH ratio is a hallmark of inflammation and cellular stress in many immune cells[72], the attenuation of these cellular processes in Orai1 cKO astrocytes indicates that blockade of Orai1 activity likely dampens inflammation by diminishing stimulus-evoked astrocyte metabolism. Increased cellular AMP in the cKO cells could also stimulate the AMP-activated protein kinase (AMPK), a central regulator of energy homeostasis and whose agonism is well-known to mitigate metabolic disorders and inflammation[73]. Thus, mitigation of inflammation observed in the cKO mice may in part be related to not only the suppression of neuroinflammatory pathways but also induction of protective signaling cascades.

It is known that intense systemic inflammation following infection or injury exacerbates symptoms of depression because of infiltrating cytokines and immune cells into the brain[6,7]. Several lines of evidence indicate that astrocyte Orai1 channels have a role in the CNS response that follows peripheral inflammation. First, our 2-photon Ca²⁺ imaging measurements indicated that peripheral inflammation elevates astrocyte Ca²⁺ signaling, especially in the distal and proximal processes of hippocampal astrocytes where many of the known critical functions of astrocytes are localized[1]. Second, analysis of inflammatory markers in the hippocampus indicated that following LPS-evoked neuroinflammation, expression of the glial markers GFAP and IBA1, the complementary cascade member C3, IL-6, and IL-1α were reduced in Orai1 cKO mice. Because NFκB-mediated C3 production and its release from astrocytes activates complement receptors on microglia to amplify cascades of inflammation[74], a reduction of C3 expression in Orai1 KO astrocytes would also diminish microglial activation following peripheral LPS stimulation, in line with decreased IBA1 labeling observed in Orai1 cKO mice. Third, conditional deletion of Orai1 in astrocytes mitigated CNS inflammation-evoked depression-like behaviors in the LPS model of neuroinflammation. Specifically, measures of anhedonia as assessed by intake of sugar water, and helplessness as assessed by immobility on Forced Swim and Tail Suspension tests were unaffected in Orai1 cKO mice following LPS challenge, in contrast to increased anhedonia and helplessness observed in WT mice. Together, these results indicate that Orai1-mediated stimulation of astrocyte metabolism and gene expression drives brain inflammation following systemic inflammatory challenge, establishing astrocytic Orai1 channels as an important link between peripheral stress and brain's inflammatory response. The mitigation of depression-like behaviors in Orai1 cKO mice following LPS challenge is consistent with growing evidence indicating that anti-inflammatory therapeutic agents mitigate major depression in some patients[10], and suggests that targeting astrocyte-mediated inflammation may aid the quest for developing new treatments for inflammatory diseases of the brain.

In the cascade of events linking peripheral inflammation to astrocyte activation, the upstream activators of Orai1 signaling in astrocytes include not only thrombin but also purinergic agonists such as ATP which stimulate P2Y and P2X receptors[23]. There is extensive evidence linking purinergic receptor activation, especially P2X7 and P2Y receptors with astrocyte reactivity and neuroinflammation[9]. Once activated, astrocyte Orai1 channels would be expected to not only directly stimulate inflammatory cytokines but also cause more ATP release from astrocytes[23], which would further activate P2X and P2Y receptors in nearby astrocytes and microglia in a feedforward manner to orchestrate runaway inflammation.

Several mechanisms may underlie the mitigation of LPS-induced behavioral depression in male Orai1 cKO mice. Inflammatory cytokines such as TNFα, IL-6, and MCP-1 are known to modulate neuronal excitability[60,61]. Thus, the reduction of these cytokines in Orai1 cKO mice predicts that LPS-induced modulation of synaptic transmission would be differentially affected in WT and Orai1 cKO mice. In line with this prediction, analysis of synaptic currents in LPS-exposed mice revealed strong suppression of sIPSC activity (both amplitude and frequency) in the Orai1 cKO mice which would be expected to have strong effects on circuit-level excitatory/inhibitory (E/I) synaptic ratios in the brain. Notably, the decrease in inhibitory synaptic transmission is analogous to the reduction in sIPSCs previously seen in CA1 hippocampal neurons by the rapidly acting antidepressants, ketamine and scopolamine[64,75]. In the case of ketamine and scopolamine, their antidepressant effects are thought to arise from disinhibition induced by suppression of GABAergic neurotransmission, effectively elevating the overall E/I synaptic ratio in the brain to relieve behavioral depression behaviors[64,75]. We postulate that the strong suppression of sIPSCs in astrocyte Orai1 cKO mice may relieve depression-like behaviors through a similar effect, raising the possibility that therapeutically

targeting astrocytic Orai1 channels may offer a path for mitigating depression.

What is the underlying mechanism of the observed decrease in sIPSC activity in Orai1 cKO mice? It is well-known that astrocytes sustain the activity of inhibitory interneurons and synaptic inhibition in the hippocampus via both phasic and tonic mechanisms[76] and this activation of interneurons is mediated, at least in part, through Orai1-dependent release of ATP from astrocytes[23]. Given that ablation of Orai1 blocks LPS-evoked increases in astrocyte $Ca^{2+}$ signaling (Fig. 6), diminished activity of Orai1 cKO astrocytes in LPS-exposed mice would be expected to reduce the release of astrocyte-derived ATP and other neuroactive agents to reduce GABAergic neuronal activity. Recent observations additionally indicate that reactive astrocytes with impaired glutamate-glutamine metabolism suppress inhibitory synaptic currents[77]. Because Orai1 cKO astrocytes show reductions in glutamate-glutamine metabolism, this phenotype could also contribute to the suppression of interneuron activity in the cKO mice independently of (or in addition to) the loss of $Ca^{2+}$-dependent gliotransmitter release. Additionally, LPS-driven activation of microglia and macrophages, which would be expected to occur in astrocyte cKO mice, is known to increase levels of IL-1β, a cytokine that elicits strong inhibition of GABAergic neurotransmission[63,78,79]. In WT neurons, the suppression may be partially counterbalanced by Orai1-dependent potentiation of interneurons by astrocytes[23]. However, because this potentiation is lost in the cKO mice, IL-1β-mediated GABAergic defects may be more robustly revealed in the Orai1 cKO mice. Thus, a multitude of effects may contribute to reduced GABAergic tone in the Orai1 cKO mice to relieve depression-like behaviors.

Our study has limitations. One issue is that the RNA-seq analysis was carried out in primary cultures of astrocytes. Because the transcriptional profile of astrocytes in vivo differs from astrocytes in culture[80], the specific contributions of Orai1 for gene expression in vivo may differ from the genes identified here. However, our in vivo analysis revealed that numerous inflammation markers including GFAP, IBA1, IL-6, IL-1α, and C3 were also downregulated in the brains of LPS-challenged Orai1 cKO mice, indicating that these inflammatory pathways are broadly impacted by Orai1 signaling both in vivo and in vitro. A second limitation is that our behavioral analysis exclusively focused on inflammation-evoked depression-like behaviors in mice. Because it is hard to extrapolate behaviors observed in mice to humans, we caution that extrapolation of our conclusions to broader depression syndromes such as major depression is not justified and requires further work. A final point is that it is unclear to what extent the astrocyte Orai1-regulation of inflammation and depression is similar between male and female mice. In our experiments, we did not note any in vitro differences in regulation of gene expression between astrocytes from male or female neonatal mice (Fig. 2C, Supplementary Fig. 4C, H and Supplementary Fig. 6). Moreover, in vivo, adult mice of both sexes showed upregulation of astrocyte and microglial activation markers following LPS challenge in the hippocampus. Ablation of Orai1 mitigated the upregulation of GFAP and IBA1 in both sexes. However, in the behavioral tests, females failed to show behaviors associated with depression-like behaviors following LPS challenge in line with previous work showing that in rodent models of neuroinflammation-induced depression, females are typically unaffected[68,70]. This is believed to be likely due to the strong protective effect of BDNF in female rodents[69,71]. Accordingly, deletion of one allele (heterozygous BDNF mutant female mice) suffices to reveal depression-like phenotypes in response to LPS[71]. More studies are needed to address the cellular mechanisms of these sex-differences, but our findings showing that ablation of Orai1 mitigates inflammatory markers in both sexes strongly suggests that targeting astrocytic Orai1 may offer a path for mitigating brain inflammation in both sexes.

## Methods

### Transgenic mice

All mice (C57BL/6) were cared for in accordance with institutional guidelines and the Guide for the Care and Use of Laboratory Animals. The research protocol for this work (protocol number IS00001817) was approved by the Northwestern University, Feinberg School of Medicine Institutional Animal Care and Use Committee. Animals were group-housed under standard housing conditions (12:12-hour light/dark cycle with lights on at 7:00 a.m., humidity between 30% and 70%, temperatures of 20° to 22 °C, and with ad libitum access to water and food), in a sterile ventilated facility. Littermates were randomly assigned to treatment or control groups and male and female mice were used in approximately equal numbers.

We used two approaches for Orai1 deletion in astrocytes. In one approach used for in vitro and brain slice experiments, Orai1$^{fl/fl}$ mice and Orai1$^{fl/+}$ were crossed with *mGFAP-Cre* mice (0l2887 from the Jackson Laboratory) to yield Orai1$^{fl/fl\ GFAP-Cre}$ mice as previously described[23]. In this line, Cre recombinase is controlled by a mouse GFAP regulatory sequence targeted to postnatal astrocytes. For all in vivo experiments and the RNA-seq analysis, astrocyte-specific deletion of Orai1 was accomplished by crossing Orai1$^{fl/fl}$ mice with *Aldh1l1-Cre/ERT2* mice (031008 from the Jackson Laboratory)[26] to yield Orai1$^{fl/fl\ Aldh1l1-Cre/ERT2}$ mice. Orai1$^{fl/fl\ Aldh1l1-Cre/ERT2}$ mice from this cross were born at Mendelian ratios and did not differ from WT mice in terms of weight, litter size, or gross mobility. Deletion of Orai1 with this inducible *Cre* line was accomplished using tamoxifen in accordance with the standard Jackson Laboratory protocol. Adult mice (8–11 weeks) were injected with 75 mg/kg tamoxifen dissolved in corn oil for 5 consecutive days. After the injections, we waited for 10 days before carrying out subsequent behavioral assays or immunohistochemistry.

For the inflammation and behavior studies, following the 10-day waiting period, mice (10–13 weeks old) were IP injected with 1 mg/kg Lipopolysaccharides from Escherichia coli O111:B4 (LPS) (Sigma; L4391) dissolved in sterile PBS or an equivalent volume of sterile PBS. Mice were monitored for signs of extreme distress for 24 h after injection with LPS.

### Primary astrocyte cultures

Stellate-like astrocytes were cultured from neonatal (P2-P5) mice using methods adapted from the AWESAM protocol with minor modifications[27]. Hippocampi were dissected and meninges were removed under a dissection microscope in 4 °C dissection medium [10 mM HEPES in Hanks' balanced salt solution (HBSS)]. The tissue was minced and dissociated in a trypsinization solution (0.25% trypsin; 100 μg/mL DNAse; in HBSS) for 15 min in a 37 °C water bath. Tissue was washed twice with HBSS and dissociated gently by trituration in culture media (Dulbecco's modified Eagle's medium with 10% fetal bovine serum and 1% penicillin-streptomycin solution). Dissociated cells were filtered through a 70-μm strainer to collect cell suspension and initially expanded in 25-mm² tissue culture flasks with 10 ml of medium. Half of the medium was exchanged every 3 to 4 days, and microglia were removed by forcefully shaking by hand for 15 to 30 s before each medium change. After 7 days, the medium was completely removed from the cells and exchanged with preheated 0.05% trypsin-EDTA. After ~5 min in the incubator, a culture medium was added to inactivate the trypsin, and cells were collected and centrifuged for 5 min. The supernatant was removed, and the cell pellet was then resuspended in a serum-free NB+ culture media consisting of Neurobasal media with heparin-binding epidermal growth factor (HBEGF) (5 ng/ml), 2% B-27 supplement, 1% GlutaMAX, and 1% penicillin-streptomycin solution. Cells were plated on poly-L-lysine-coated glass-bottom dishes (MatTek, 14-mm diameter, 5000 to 10,000 cells per coverslip) or 12-well plates (~50,000 cells per well). These serum-free astrocytes were maintained in the incubator and used for experiments after 7 days of

culture and within 14–21 days of this plating. Half of the NB+ medium was exchanged with fresh medium every 7 days.

Orai1*fl/fl GFAP-cre* and Orai1*fl/fl Aldh1l1-Cre/ERT2* astrocytes exposed to 4-OH tamoxifen exhibit near complete deletion of Orai1 by RT-qPCR. In astrocytes cultured from Orai1*fl/fl GFAP* mice RNAseq gene counts revealed that Orai1 expression was diminished by 72% relative to Orai1*fl/fl* astrocytes (Supplementary Fig. 4). Note that this is likely an underestimate of the deletion efficiency because the vast majority of the mRNA transcripts detected by RNAseq in the cKO cells were derived from exon1, which is not excised by the Cre. The coding region, in exon2, shows a more dramatic decrease compared to WT transcripts (Supplementary Fig. 4d). For inducible deletion of Orai1 in culture, astrocytes from Orai1*fl/fl Aldh1l1-Cre/ERT2* hippocampal cultures were treated with 1 μM 4-OH tamoxifen (dissolved in ethanol, #H7904, Sigma-Aldrich, USA) added to the culture flask medium on days 2 and 4. Medium containing tamoxifen was removed on day 7 when DMEM + FBS was replaced with NB+ medium.

## Immunostaining

Astrocytes plated on coverslips were fixed with 10% formalin in PBS for 15 min at 4 °C. Cells were washed with PBS, blocked in 5% goat serum in PBS for 1 h. Cells were incubated with primary antibodies at 1:500 for GFAP (rabbit polyclonal, Invitrogen PA1-10019) and 1:200 for Orai1 (Abcam, 266.1, ab175040) at 4 °C overnight, followed by secondary antibody tagged to Alexa Fluor 488 or Alexa Fluor 594 at 1:500 for 1 h. Nuclei were labeled with 4′,6-diamidino-2-phenylindole.

## Wide-field fura-2 Ca²⁺ imaging

Astrocytes grown on glass-bottom dishes were loaded with Fura-2 by incubating cells in 2 mM Fura-2–AM (Invitrogen) in NB+ medium for 35 min at 37 °C. Fura-2–containing medium was washed off, and cells were incubated in the medium for an additional 5 to 10 min before imaging. All experiments were performed at room temperature. Single-cell $[Ca^{2+}]_i$ measurements were performed as described previously[23]. Image acquisition and analysis were performed using SlideBook (Denver, CO). Dishes were mounted on the stage of an Olympus IX71 inverted microscope, and images were acquired every 6 s at excitation wavelengths of 340 and 380 nm and an emission wavelength of 510 nm.

The standard Ringer's solution used for wide-field Ca²⁺ imaging studies contained the following: 155 mM NaCl, 4.5 mM KCl, 10 mM D-glucose, 5 mM Hepes, 1 mM MgCl₂, and 2 mM CaCl₂. The Ca²⁺-free Ringer's solution contained 3 mM MgCl₂, 1 mM EGTA (Sigma-Aldrich), and no added CaCl₂. pH was adjusted to 7.4 with 1 N NaOH. Stock solutions of thapsigargin (TG) were dissolved in dimethyl sulfoxide and used at the indicated concentrations.

For data analysis, ROIs were drawn around single cells, the background was subtracted, and F340/F380 ratios were calculated for each time point. $[Ca^{2+}]_i$ was estimated from the F340/F380 ratio using the standard equation:

$$[Ca^{2+}]i = \beta K_d (R - R_{min})/(R_{max} - R) \tag{1}$$

where $R$ is the $F_{340}/F_{380}$ fluorescence ratio and values of $R_{min}$ and $R_{max}$ were determined from an in vitro calibration of Fura-2 pentapotassium salt. $\beta$ was determined from the $F_{min}/F_{max}$ ratio at 380 nm and $K_d$ is the apparent dissociation constant of Fura-2 binding to Ca²⁺ (135 nM). For each cell, the rate of SOCE ($\Delta[Ca^{2+}]_i/\Delta t$) was calculated from the slope of a line fitted to three points (12 s) after the readdition of 2 mM Ca²⁺ Ringer's solution. Total calcium release and entry were assessed using the area under the curve (A.U.C.) for three minutes following application of TG (store release) or for five minutes following calcium readdition following store depletion (Ca²⁺ entry).

## RNA-sequencing

**RNA isolation.** Astrocytes from littermates (either Orai1*fl/fl* or Orai1*fl/fl: GFAP-Cre*) were cultured as described above for three weeks and then treated with 0.2 μM TG + 50 nM PDBu or equivalent volume of DMSO for 6 h. After treatment, RNA isolation was performed from cultured cells using the QIAGEN RNeasy kit with gDNA extraction columns. Total RNA integrity and quantity were measured using Agilent 4200 Tapestation automated electrophoresis platform using their proprietary high Sensitivity RNA ScreenTape System (Agilent Technologies).

**Library preparation.** mRNA was isolated from purified 50 ng total RNA using oligo-dT beads (New England Biolabs, Inc). NEBNext Ultra™ RNA kit was used for full-length cDNA synthesis and library preparation. Libraries were pooled, denatured and diluted, resulting in a 1.8 pM DNA solution. PhiX control was spiked at 1%.

**Bulk RNA-sequencing.** Libraries were sequenced on an Illumina NextSeq 500 instrument (Illumina Inc) using NextSeq 500 High Output reagent kit (Illumina Inc) (1 × 75 cycles) with a target read depth of approximately (5–10) million aligned reads per sample. FASTQ files were generated using bcl2fastq 2.19.1 (Illumina). To facilitate reproducible analysis, samples were processed using the publicly available nf-core/RNA-seq pipeline version 3.4 implemented using nextflow 21.10.5.5658 and singularity 3.8.12 with the public *nu_genomics* configuration. The STAR/Salmon method was used for transcriptome alignment and gene-level count assignment. The GRCm38 genome (Ensembl release 81) was used as a reference. The resulting length-scaled DESeq2 object was then imported directly for downstream analysis.

**Differential expression gene (DEG) analysis.** Analysis was done using custom scripts in R ver 4.1.1 using the DESeq2 1.34.0 framework[81]. For differential expression analysis (DEA), a single combined factor encompassing genotype and treatment was used to model expression. A "local" model of gene dispersion was used, as this better-fit dispersion trends without obvious overfitting, and alpha was set to 0.05 for all comparisons; otherwise, default settings were used (see code for details). For group-wise comparisons, gene expression was compared using Wald tests. For identifying variable genes across all groups (as in heatmaps of variable gene expression), a likelihood ratio test (LRT) was used, with the combined factor as the "full" model, and the intercept as the "reduced" model (-1).

**Gene enrichment analysis.** Standard gene ontology (GO) enrichment analysis was done using the topGO 2.46.0 package. Enrichment was determined using Fisher's exact test using the "classicCount" implementation. The gene "universe" was defined as all genes for which at least one count was detected across all samples. GO terms were defined using the org.Mm.eg.db 3.14.0 package. Resultant p-values were manually FDR-corrected. For gene-set-enrichment analysis (GSEA), the fgsea 1.20.0 package was used[82]. Both the "Reactome" and "Hallmark" gene set lists were downloaded from the Molecular Signatures Database (MSigDB) 7.5.1 at http://www.gsea-msigdb.org/gsea/downloads.jsp. Mouse orthologs for all genes were then determined using the Ensembl database release 105 through biomaRt 2.50.1 using the hsapiens_homolog_associated_gene_name parameter. Enrichment analysis was then performed for all gene sets simultaneously using the "fgseaMultilevel" method using gene-level Wald statistics as rankings and default parameters.

**Statistical analysis for RNA-seq.** Statistical analysis was performed using base R 4.1.1 with the aid of tidyverse version 1.3.1. For all comparisons, normality was first assessed using a Shapiro–Wilk test and manual examination of distributions. For parameters that exhibited a

**Table 1 | Methods: Primer sequences for two-step qPCR**

| Gene | Forward | Reverse | PrimerBank ID/other source |
|------|---------|---------|----------------------------|
| IL-6 | CTGCAAGAGACTTCCATCCAG | AGTGGGTATAGACAGGTCTGTTGG | 13624310c1 |
| IL-1a | CGAAGACTACAGTTCTGCCATT | GACGTTTCAGAGGTTCTCAGAG | 52669a1 |
| IL-33 | TCCAACTCCAAGATTTCCCCG | CATGCAGTAGACATGGCAGAA | 19527000a1 |
| MIP1a | TTCTCTGTACCATGACACTCTGC | CGTGGAATCTTCCGGCTGTAG | 6755432a1 |
| MCP1 | TTAAAAACCTGGATCGGAACCAA | GCATTAGCTTCAGATTTACGGGT | 6755430a1 |
| TNFa | CCCTCACACTCAGATCATCTTCT | GCTACGACGTGGGCTACAG | 7305585a1 |
| IL1RA | AGAACTCGCCTGTGGTTTTG | TTCCAAAGTGAGCTCGGTAAA | https://doi.org/10.1038/mtna.2012.58 |
| TGFB2 | GCAGGGGCAGTGTAAAC | GCAGGGGCAGTGTAAAC | 15029686a |
| IL18BP | CCTACTTCAGCATCCTCTACTGG | AGGGTTTCTTGAGAAGGGGAC | 6754314a1 |
| IL-10 | GCTCTTACTGACTGGCATGAG | CGCAGCTCTAGGAGCATGTG | 6754318a1 |

clear lack of normality, nonparametric tests were used. In cases of multiple testing, P values were corrected using FDR correction. Adjusted P values < 0.05 were considered significant. Two-sided statistical tests were performed in all cases. Bar graphs show the mean ± SEM of the individual values. Tests with more than two groups were done using one-way ANOVA followed by the indicated posthoc test (Tukey typically or as indicated). For comparison of our TG+PDBu gene expression results to the DEGs previously seen for astrocytes following peripheral LPS stimulation, we used the LPS datasets described by Srinivasan et al[30].

**Data visualization.** Plotting was performed in Fig. 2 and Fig. S2 using ggplot2 3.3.5. Comparisons were added using ggsignif 0.6.3. Heatmaps were generated using pheatmap 1.0.12. For all box plots, box limits represent the interquartile range (IQR) with a center line at the median. Whiskers represent the largest point within 1.5× IQR. All points are overlaid. Plots generated in R were exported using Cairo 1.5–12.2 and edited in Adobe Illustrator.

**Metabolomics**

Primary cultured astrocytes (*Orai1*[fl/-] or *Orai1*[fl/fl GFAP-Cre] (cKO)) were stimulated with 0.2 μM TG + 50 nM PDBu or DMSO for 6 h. Cells were rinsed with ice-cold saline twice and then scraped in chilled 80% methanol (−80 C) on dry ice. Lysate was chilled at −80 °C for 5 min and then vortexed at room temperature for 60 s; this cycle has repeated a total of three times and lysate was stored at −80 °C overnight and then centrifuged at 20,000 x *g* for 15 min at 4 °C. Extraction solution was then removed and dried with a SpeedVac. The remaining protein pellet was dissolved in a buffer containing 8 M urea then diluted for BCA analysis and used to calculate the injection amount. 60% acetonitrile was added to the tube for reconstitution followed by overtaxing for 30 s. The sample solution was then centrifuged for 30 min @ 20,000 g, 4 °C. Supernatant was collected for LCMS analysis.

**Metabolomics analysis.** Samples were analyzed by High-Performance Liquid Chromatography and High-Resolution Mass Spectrometry and Tandem Mass Spectrometry (HPLC-MS/MS). Specifically, the system consisted of a Thermo Q-Exactive in line with an electrospray source and an Ultimate3000 (Thermo) series HPLC consisting of a binary pump, degasser, and auto-sampler outfitted with a Xbridge Amide column (Waters; dimensions of 3.0 mm × 100 mm and a 3.5 μm particle size). In positive/negative polarity switching mode, an *m/z* scan range from 60 to 900 was chosen and MS1 data was collected at a resolution of 70,000. The automatic gain control (AGC) target was set at $1 × 10^6$ and the maximum injection time was 200 ms. The top 5 precursor ions were subsequently fragmented, in a data-dependent manner, using the higher energy collisional dissociation (HCD) cell set to 30% normalized collision energy in MS2 at a resolution power of 17,500. Besides matching m/z, metabolites are identified by matching either retention time with analytical standards and/or MS2 fragmentation pattern. Data acquisition and analysis were carried out by Xcalibur 4.1 software and Tracefinder 4.1 software, respectively (both from Thermo Fisher Scientific). Quantitative Enrichment Analysis was performed at metaboanalyst.ca. Visualization was performed in OriginLab (OriginPro, Version 2022) and Matlab (version 2023a) and edited in Adobe Illustrator. Metabolomics data have been deposited at https://github.com/PrakriyaLab/Astrocyte2022.

**Two-step RT-qPCR**

RNA from cultured hippocampal astrocytes (plated in 12-well plates) was isolated using the RNeasy kit (QIAGEN). Briefly, cells were lysed in the wells for 1–3 min with the RNeasy lysis buffer. Cell lysis solution was then passed through a gDNA eliminator column to eliminate genomic DNA, and RNA was acquired following the RNeasy protocol. cDNA libraries were made from the isolated RNA using High-Capacity RNA-to-cDNA™ Kit (Applied Biosystems). Primers for qPCR were acquired from Integrated DNA Technologies. RT-qPCR was performed on a Bio-Rad CFX connect real-time PCR system and analyzed using BioRad CFX Maestro software. Sequences of primers can be found in Table 1.

**Plasmids and transfection**

Astrocytes were transfected using Lipofectamine 2000 (Invitrogen) according to the manufacturer's instructions. pSIRV-NFAT-eGFP was obtained from Dr. Mark Del'Aqua (University of Colorado). The pGL3-NFAT-Luc and pRL-Tk-Luc vectors were obtained from Dr. Richard Lewis (Stanford University). Experiments were performed 24–48 h after cell transfection.

**Transcription factor luciferase assays**

NFAT and NFKB activity in astrocytes was assayed using luciferase activity of an NFAT-dependent luciferase construct, pGL3-NFAT, or NFκB -dependent luciferase construct, pGL3-NFκB. The NFAT- or NFKB-driven firefly luciferase activity was normalized to the constitutively active renilla luciferase (pRL-Tk-Luc) activity to correct for variations in cell density and transfection efficiency. Astrocytes were transfected with the firefly luciferase and pRL-Tk-Luc vectors at a ratio of 15:1 24–48 h before experiments. Cells were stimulated with 0.2 μm thapsigargin (TG) and 50 nM PDBu or 2 U/mL Thrombin for 4 h. Cells were lysed and luciferase activity was measured using the Dual Luciferase Reporter Assay kit (Promega) and a single tube luminometer (Berthold Instruments).

**Enzyme-linked immunosorbent assays (ELISAs)**

In vitro ELISA assays were performed using cell supernatants from AWESAM astrocytes cultured in 12-well plates. For assays performed on adult mouse brain tissue, RIPA buffer was used to isolate protein from the brain lysate. Briefly, adult mice were anaesthetized with isoflurane and then perfused with ice-cold saline. Brain hemispheres were

**Table 2 | Methods: Kits used for ELISA**

| Kit Name | Manufacturer | Catalog number |
|---|---|---|
| Mouse IL-1a ELISA kit (tissue) | Abcam | Cat# ab199076-196TESTS |
| Mouse IL-6 ELISA kit (Tissue) | RayBiotech | ELM-IL6-CL-1 |
| Mouse IL-6 ELISA kit (supernatant) | RayBiotech | ELM-IL6-1 |
| PGE-2 ELISA kit | Cayman | KGE004B |
| Mouse MCP-1 ELISA kit (tissue) | Abcam | Cat# ab208979 |
| Thrombin Mouse Elisa kit | Abcam | Cat# ab230933 |
| SDF1α Mouse ELISA kit | Abcam | Cat# ab100741 |
| SDF1α Mouse ELISA kit | Abcam | Cat# ab100741 |
| Mouse IL-1a kit (supernatant) | RayBiotech | ELM-IL1a-1 |
| Mouse MIP1a kit (supernatant) | RayBiotech | ELM-MIP1a-1 |

flash-frozen on dry ice and stored at −80 °C. Hippocampi and associated cortex were dissected out, minced, triturated in ice-cold RIPA buffer, and shaken for two hours at 4 °C. After shaking, the mixture was centrifuged for 20 min at 4 °C. Supernatant was used for ELISAs and for protein quantification by BCA. Cytokine concentrations were normalized to average protein in lysates. ELISA kit information can be found in Table 2. Manufacturer protocols were followed in all cases.

### Administration of LPS to adult mice

For the in vivo inflammation and behavior studies, following the 10-day waiting period, Orai1$^{fl/fl}$ or Orai1$^{fl/f\ Aldh1l1-Cre/ERT2}$ mice (10–13 weeks old) were IP injected with 1 mg/kg Lipopolysaccharides from Escherichia coli O111:B4 (LPS) (Sigma; L4391) dissolved in sterile PBS or an equivalent volume of sterile PBS. Littermates were randomly sorted into treatment or control groups. Mice were monitored for signs of extreme distress for 6 h after injection with LPS. Behavioral assays were performed 24 h after LPS/saline injection and immunohistochemical assays were performed on tissue from mice sacrificed 48 h post LPS/saline injection.

### Immunohistochemistry

Mice were anaesthetized with isoflurane and perfused with ice-cold saline to clear the brain of blood. Brain hemispheres were dissected and placed in ice-cold 10% formalin and fixed overnight. The next day, the hemispheres were rinsed with PBS and placed in a preserving solution (30% sucrose and 0.1% sodium azide in PBS). Tissue was cut serially into 30 μm thick sections and stored floating at −20 °C in a freezing solution (30% sucrose, 30% ethylene glycol in 0.1 M phosphate buffer). Before staining, slices were rinsed with tris buffered saline (TBS) and incubated for 20 min at 80 °C in 0.1 M sodium citrate buffer pH 9 for antigen retrieval. Sections were allowed to equilibrate to room temperature and then rinsed again in TBS. Sections were incubated for 1 h in 16 mM glycine (dissolved in TBS) and rinsed again. For GFAP (Thermofisher Cat #14-9892-82)/IBA1 (Thermofisher Cat #PA5-27436)/C3 (Thermofisher Cat #PA5-21349), sections were blocked in 5% goat serum in TBS with 0.25% Triton-X 100 for 1-2 h and washed in rinsing solution (1% BSA with 0.25% Triton-X 100 in TBS). Sections were incubated in the primary antibody solution (all 1:300) overnight at 4 °C with shaking.

For Orai1 staining, the protocol was adapted to reduce membrane disruption. Triton-X 100 was excluded until after sections were incubated with the primary Orai1 antibody (1:200; Abcam 266.1, ab175040). After the overnight incubation, slices were rinsed and incubated overnight with GFAP antibody (1:300; rabbit polyclonal, Invitrogen PA1-10019) at 4 °C with shaking. After incubation with primary antibodies, slices were incubated with secondary antibodies tagged to Alexa Fluor 488 and Alexa Fluor 594 at 1:500 for 2 h at room temperature in the dark. Nuclei were labeled with 4′,6-diamidino-2-phenylindole (DAPI). Sections were mounted on charged glass slides with Prolong Gold (ThermoFisher). Confocal imaging of the sections was performed on a Nikon A1R upright confocal microscope using a 25X Nikon CFI APO LWD objective. Image analysis was performed using the Nikon Elements software.

Illustrative images in this manuscript are background-subtracted in Nikon Elements software. Image J was used for any further adjustment and to export to figures. Any brightness adjustment necessary for visualization was applied equally to all images in a figure. For quantification of % area, images were background subtracted in Nikon Elements using a background region of interest. A consistent threshold for each color (stain) was determined, and the area of immunofluorescence above that threshold was divided by the area of the region of interest for each image. GFAP, IBA1, and C3 areas were quantified independently. For quantification of Orai1+ astrocytes in Orai1$^{fl/fl\ Aldh1l1-Cre/ERT2}$ (cKO) and Orai1$^{fl/fl}$ (WT) mice, sections were imaged in 1 μm-apart z-stack to obtain 5 μm-thick sections. In the max-IP images, GFAP+ cells outside of the pyramidal layer (where they could be clearly differentiated from other cells) were identified and ROIs were drawn to include the nucleus + 5 μm. ROIs with Orai1 staining above background were counted as Orai1+ astrocytes while ROIs without Orai1 staining were counted as Orai1- astrocytes. The percentage was determined from 56–311 cells/mouse.

### Brain slice preparation

For electrophysiology, hippocampal slices (300–350 μm) were cut from the brains of 1.5–3-week-old Orai1$^{fl/fl}$ or Orai1$^{fl/fl\ GFAP-cre}$ mice. For 2PLSM, the same protocol was used except that hippocampal slices (250 μm) were cut from brains of 10–12-week-old mice. Mice were anesthetized with isoflurane and promptly decapitated. Brain dissections were performed in ice-cold sucrose artificial cerebrospinal fluid (ACSF), containing (in mM): 83 NaCl, 2.5 KCl, 26.2 NaHCO₃, 1 NaH₂PO₄, 0.5 CaCl₂, 3.3 MgCl₂, 22 glucose, and 72 sucrose. Coronal brain slices were acquired at 300–350 μm thickness using the Compresstome VF-200-0Z (Precisionary Instruments, MA) and left to incubate for 30 min in a chamber with recovery ACSF, which was continuously oxygenated with 95% O2/5% CO2 and kept at 37 °C. The makeup of recovery ACSF (in nM) is as follows: 130 NaCl, 3.5 KCl, 1.25 NaH₂PO₄, 24 NaHCO₃, 2 CaCl₂, 1 MgCl₂, and 10 D-Glucose. After 30 min in physiological temperature, the chamber and slices were transferred and left to equilibrate at room temperature for at least 30 min. After, brain slices were used for electrophysiology recordings for up to 6–8 h. Brain slices were continuously oxygenated under all conditions.

### Slice electrophysiology

A total of 16 mice (postnatal days 10–21) were used for electrophysiology experiments, with approximately equal numbers of males and females. All whole-cell electrophysiology recordings were acquired in voltage-clamp mode using the Axopatch 200B amplifier, 1550B digitizer, and pClamp 11.1 (Molecular Devices, CA). All recordings were acquired with a 2 kHz low-pass filter and sampled at 10 kHz. Data were analyzed with Clampfit 11.1 (Molecular Devices, CA) and MiniAnalysis (Synaptosoft). Brain slices were gently transferred to a RC-26GLP recording chamber (Warner Instruments, CT) and constantly perfused with oxygenated recovery ACSF at -1 ml/min. Hippocampal lamina and the CA1 pyramidal layer were visualized and identified with the Olympus BX51WI microscope (Olympus, Japan). A total of 35 neurons were recorded for these experiments.

Borosilicate glass (Sutter Instruments, CA) was pulled with the P-97 Micropipette Puller (Sutter Instruments, CA), yielding recording electrodes with a final tip resistance of 4–7 MΩ. Electrodes were filled with an intracellular solution containing (in mM): 125 CsMeSO₃, 16 KHCO₃, 10 QX-314, 4 MgATP, and 0.3 Na2GTP. After achieving whole-cell break-in into the hippocampal CA1 pyramidal neurons, spontaneous excitatory postsynaptic currents and inhibitory postsynaptic currents (sEPSCs and sIPSCs, respectively) were recorded while the neuron was clamped at reversal potentials for AMPA and GABAA at

0 mV and −70 mV, respectively. Multiple excitatory and inhibitory current traces were collected, with each trace lasting for 1 min. Current amplitudes were determined by first choosing sEPSCs and sIPSCs that were above a set threshold current in MiniAnalysis, which were then averaged during the duration of the chosen sweep. Frequency was determined by dividing the number of currents by 60 s in a chosen sweep.

## Two laser scanning microscopy (2PLSM)

A total of 12 mice (10–12 weeks old) were used for 2PLSM $Ca^{2+}$ imaging studies, with approximately equal numbers of males and females. Mice (8–10 weeks old) were virally injected with the AAV5 gfapABC1D-cyto-GCaMP6f (Addgene plasmid no. 52925) into the hippocampus at a titer of $1.4 \times 10^{13}$ viral genomes/ml at a volume of 0.5 µl over 5 min as previously described[23]. Two weeks later, mice were intraperitoneally injected with either saline or 1 mg/mL LPS. The following day, mice were anesthetized with isoflurane, and brain slices were obtained as described above. GCaMP6f signals in brain slices were imaged using a Nikon A1R-MP Multiphoton Microscope with a Chameleon Vision titanium sapphire laser and 25x Nikon objective lens using the Nikon Elements software (Nikon Instruments Inc, Melville, NY) as previously described[23]. The hippocampus was identified in 750 nm light at low-magnification and the stratum radiatum of the CA1 region was localized. Astrocytes expressing GCaMP6f were selected in this region at depths of typically 50 to 120 µm below the slice surface and imaged using 950 nm light (10% laser power) at Nyquist resolution. Data were acquired at the rate of 1 frame/sec at room temperature. During the experiment, cells were continuously oxygenated by gently bubbling 95% O2 in the imaging chamber. Following image acquisition, regions of interest (ROIs) for the background (region not expressing GCaMP6f), soma, primary processes, and distal branches were drawn on the maximum intensity projection (Max-IP) images from the 180 s time-lapse. Intensity values from these background-subtracted ROIs were exported from Nikon Elements software and the fluorescence changes ($\Delta F/F_O$) were computed for each ROI, using the mean of the first 20 s set to $F_O$. Traces were analyzed using the "Peak Analyzer: Find Peaks" function on OriginLab (OriginPro, Version 2022. OriginLab Corporation, Northampton, MA, USA). A moving baseline with asymmetric least squares smoothing was used, and amplitude peaks in the $\Delta F/F_O$ traces were determined from local maxima that exceeded a threshold of 20% over the local baseline. For each ROI, the frequency and amplitude for all the peaks was averaged and calculated for each compartment and the frequency (peaks/min) was computed by dividing the number of peaks in the experiment by the duration of each trace (3 min of imaging).

## Mouse behavioral analysis

All behavioral tests were performed on group-housed 10–13 week-old adult mice. $Orai1^{fl/fl\ Aldh1l1-Cre/ERT2}$ and $Orai1^{fl/fl}$ mice were randomly assigned to be administered LPS (1 mg/kg) or saline. The same handler injected the mice with tamoxifen, handled the mice during the waiting period to ensure mice were acclimated to the handler, and performed the entire series of behavioral experiments. After IP injection with LPS or saline, mice were monitored for 6 h for sickness behavior. Sickness behavior was assessed using a previously-developed metric[83]. An initial Open Field Test (OFT) was performed after 6 h to test for decreased locomotion due to LPS-induced sickness. 24 h after injection, the following behavioral tests were performed in order: Open Field (OFT), tail suspension (TST), forced-swim (FST), and sucrose-preference (SPT).). Animals were anesthetized with isofluorane and perfused with saline before decapitation and brain preservation for protein quantification and immunohistochemistry 48 h after LPS injection. Data for mice where the tracking system failed (in the open field test) or if the mouse fell off the zero maze, or where mice exhibited climbing

behavior (tail suspension test) or excessive diving (in the forced swim test) was excluded. The following tests were employed:

**Tail suspension test.** Mice were suspended by the tail from a bar 10–15 cm above a table using laboratory tape. Tape was secured ~1 cm from the tip of the mouse tail with a 3–4 cm tube placed between the mouse rump and tape to deter the mouse from climbing during the experiment. Shields were placed between mice. The mice were filmed for 6 min and then returned to their cage. Blinded scoring was performed from the acquired videos by timing the immobile bouts of each mouse during the first 6 min of tail suspension.

**Forced-swim test.** Mice were placed into 8-inch diameter cylindrical glass vessels containing water at ~25 °C. Vessels were filled to ~8-inch depth to ensure that mice could not touch the bottom of the container and shields were placed between chambers. Mice were filmed swimming/floating for 6 min and then dried and returned to their cage. Blinded scoring was performed from the acquired videos by adding together the times during which the mouse was immobile (floating - not swimming) during the final 4 min of the experiment (the exception was Supplementary Fig. 14 where we analyzed all 6 min). Greater immobility time was interpreted as greater helplessness.

**Sucrose-preference test.** Mice were placed into individual mouse cages was fitted with two bottles containing 1) 1% sucrose in tap water and 2) tap water which were randomly ordered to minimize place-preference. Mice were also provided chow *ad libitum* and left in cages overnight (16 h). Bottles were weighed before and after the test to quantify the amount of sucrose-water and water consumed in order to determine the percent sucrose-water consumed. Wild-type mice typically prefer sucrose-water to plain water. This test is a measure of anhedonia: lower % sucrose-water consumed corresponds to greater anhedonia.

**Open field test.** Mice were placed into individual chambers containing an 80 cm square box. Mice were allowed to act freely within the box for 8 min during which time they were filmed. Limelight software was used to score distance traveled by the mice and time spent in the middle (exposed) area of the box.

**Y-maze.** Mice were placed at the end of an arm in the Y-maze and allowed to move freely between the three arms for 8 min while being filmed. The order of arm entries was analyzed manually. A successful alternation was recorded for each set of three consecutive arm choices in which no repeated entries occurred. An unsuccessful alternation was recorded for each set of three consecutive arm choices in which a repeat mouse entry occurred. An arm entry was counted if the entire body of the mouse (excluding tail) entered the arm. Each mouse was analyzed until it had completed 20 possible (successful and unsuccessful) alternations. The number of successful alternations was divided by the number of possible alternations for the outcome measure. Chance level of alternation is 50%.

**Zero maze.** Mice were placed in a closed area of a zero maze and allowed to move freely for 8 min while being filmed. Limelight software was used to track mouse location and the amount of time spent in the exposed areas of the maze.

**Fear conditioning.** Associative learning was assessed with a fear conditioning protocol utilizing foot shock as the unconditioned stimulus and a tone, trace, and context as the conditioned stimulus. A computer-controlled tone/shock generator (Habitest modular system; Coulbourn, Holliston, MA) and motion monitor (FreezeFrame software; Actimetrics, Wilmette IL) were used. Mice were allowed to explore context 1 (plexiglass chamber with wire grid floor) for 150 s,

after which a 30 s tone (5 kHz, 75 kB) was played, followed by a 15 s pause (trace), followed by a foot shock (2 s; 0.7 mA; constant current). The tone-shock-trace protocol was repeated twice more with 350 s between each occurrence. 24 h later, mice were returned to context 1 and allowed to explore for 5 min, during which freezing was measured. Several hours later, mice were placed in a novel context 2 (an opaque, plastic chamber with a solid floor) and allowed to explore for 180 s before the tone was played. Freezing was measured during the first exploration phase, tone (60 s), and trace (131 s).

**Murine sepsis score analysis.** Mice were assessed for sickness behaviors as described previously[83]. Briefly, mice were assessed for coat appearance, activity, consciousness, responsiveness, eye condition, respiration rate, and respiration quality.

### Statistical analysis

Statistical details of experiments may be found in the relevant figure legends. The data analysis for RNA-seq and metabolomics analysis was done as described in the earlier sections. As indicated in each legend, n indicates mice used or cells used (with a number of mice from which cells are derived also provided). In in vitro experiments, each n is taken from at least two technical replicates. All data are expressed as means ± SEM. For datasets with two groups, statistical analysis was performed with a two-tailed $t$ test to compare between control and test conditions. Paired two-tailed t-tests were performed when the same cells were measured before and after treatment. For datasets with greater than two groups, either one-way or two-way ANOVA followed by a Tukey post hoc test was used to compare groups and genotypes as appropriate (specified in figure legends). Statistical analyses for the electrophysiological results were performed using Prism 9.4.1 (GraphPad, CA). Other statistics were performed using OriginLab (OriginPro, Version 2022). Statistical analysis was performed with a confidence level of 95%, and results with $P < 0.05$ were considered statistically significant.

### Reporting summary

Further information on research design is available in the Nature Portfolio Reporting Summary linked to this article.

## Data availability

RNA-seq data generated in this study have been deposited at https://github.com/PrakriyaLab/Astrocyte2022[84] and at https://www.ncbi.nlm.nih.gov/sra under accession code PRJNA933208 and are publicly available. Metabolomic data have been deposited at https://github.com/PrakriyaLab/Astrocyte2022. Source data for all summary graphs for all experiments (excel file) are provided with this paper. Source data are provided with this paper.

## Code availability

All original code has been deposited at https://github.com/PrakriyaLab/Astrocyte2022. Custom scripts for differential expression analysis can be found at https://github.com/NUPulmonary/utils[84]. Any additional information required to reanalyze the data reported in this paper is available from the lead contact upon request. Source data (excel files) are provided with this paper.

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

## Acknowledgements

We thank members of the Prakriya laboratory for helpful discussions and Dr. Mohamed Trebak and Dr. Gabriela Piso for their comments on the manuscript. Metabolomics services were provided by the Metabolomics Core Facility at Robert H. Lurie Comprehensive Cancer Center of Northwestern University. We thank Yuliya Politanska in the Pulmonary RNA-seq core for assistance with the preparation and processing of the RNA samples, and Dr. Mark Dell'Acqua (U of Colorado) for GFP-NFATc3. Figure schematics were developed using BioRender.com. This work was supported by NIH grants R01NS057499 and R35NS132349 to MP, P01AG049665 and R01HL158139 to GRSB, and R01MH108837 to JR. MMN was supported by the predoctoral training grant T32 AG020506. RAG was supported by a predoctoral NIH NRSA, F31AG071225.

## Author contributions

M.M.N. performed the astrocyte cytokine analysis, metabolomics, immunostaining, Ca$^{2+}$ imaging, and behavioral analysis of Orai1 KO mice. K.S.K .performed the brain slice electrophysiology and analysis of the synaptic currents. R.A.G. carried out the RNA-seq analysis of WT and Orai1 KO astrocytes under supervision from GRSB. H.A.V. was responsible for the RNA-sequencing procedures. M.E.M. performed the mouse care and genotyping, viral injections, and assisted with the fear conditioning tests. J.R. helped guide the behavioral studies and editing of the paper. M.M.N. and M.P. conceived and designed the overall project and all the experiments. M.M.N. and M.P. wrote and revised the paper.

## Competing interests

The authors declare no competing interests.
