## [Peer review file · Nature Communications]

Astrocyte reactivity and inflammation-induced depression are regulated by Orai1 calcium channelsREVIEWER COMMENTS

Reviewer #1 (Remarks to the Author):

In their report „Regulation of astrocyte-mediated brain inflammation by Orai1 channels”, Novakovic et al., describe the consequences of conditional deletion of Orai1 channels on astrocytic release of inflammatory cytokines, on altered induced gene expression with influence on cellular metabolism and on behavioral changes of altered astrocyte reactivity in the context of inflammation-evoked depression. Overall, the experiments are well performed and cover a very wide range of experimental approaches with an excellent technical quality.

In Toth et al (Sci. Signaling 2019), the Prakriya group already reported effects of genetic deletion of Orai1 in astrocytes using the GFAP-Cre as recombination driver. In this previous manuscript, the authors describe in detail effects of deletion of either Stim1 or Orai1 on gliotransmitter and ATP release from astrocytes and subsequent effects on sIPSC frequencies triggered by released ATP in CA1 neurons. Here, the increased GABAergic transmission by thrombin induced release of astrocytic ATP which then stimulated interneurons, was lost with Orai1 deletion. In addition, the Prakriya group very recently demonstrated that deletion of Orai1 from brain microglia reduces inflammatory cytokine production and leads to dysregulation in gender specific neuropathic pain behavior (Tsujikawa et al., 2023). In 2017, Vaeth et al, reported in detail how SOCE controls metabolic reprogramming in T Cells.

Given these previous findings, the overall novelty of the current manuscript, although very thorough and with clear sets of data is limited and in need of a more critical discussion of combining different findings both of the Prakriya lab as well as of other groups, in addition to a few control experiments.

1. Is the reduced ATP release seen in Orai1 deleted astrocytes (Toth et al) due to reduced release (Toth et al) or due to an overall increase in AMP/ATP ratios and reduced ATP (Fig 3 F,G, current MS) and thereby leading to a reduced vesicular ATP concentration? Same applies for the reduced glutamate release reported in Toth et al. which may be due to altered glutamine metabolism. In Toth et al., a direct role of SOCE in release of vesicles was postulated. Is this result still valid?

2. Inhibition of GABA interneurons (as seen by Toth et al) has been shown to be sufficient and necessary for an antidepressant response (Fogaca et al., Mol Psych. 2021). Is this the explanation for the behavioral phenotype of the Orai KO astrocytes, potentially independent from inflammatory aspects? What happens if stress stimuli and not inflammation are used to trigger depression?

3. Release of many pro-inflammatory cytokines usually entails activation of P2RX7 channels and indeed, acute stress induces depressive-like behavior has been shown to be mediated by astrocytic and

microglial P2X7 channels (Zhao YF et al., 2022, see also 51 citations on astrocytes, P2X7 and inflammatory cytokines). These data need to be discussed and using the available RNA-Seq data, the authors need to show in immunohistochemical staining of adult astrocytes from tamoxifen induced Orai1KO (or GFAP-Cre) if levels of P2X7, and some selected components such as pannexins, connexins, TLR receptors (or other pathways of inflammation (Fig. 2B)) are altered in vivo. If downregulated, mediators of the classic LPS triggered NLRP3 inflammasome, potentially indirectly protect mice from inflammation induced anhedonia, helplessness and despair. In spinal slices, astrocytic connexin43 hemichannels respond to LPS (Panattoni et al., Mol Brain 2021).

4. How do levels of released cytokines compare in WT and KO cells when using a classical dual activation protocol (i.e. LPS+ATP)?

5. Overall, the amounts of released cytokines are low (~50 pg/ml for IL-6 after Tg/PDBu, Fig. 4B, compared to ~3200 pg/ml after LPS (Lu, X et al, J Neuroinflamm.) or 400 pg/ml after IL-1 α /TNF α stimulation (Nakajima et al., 2022), in this later report, release of PGE2 was shown to be enhanced by KD of Orai2. In addition, Thrombin induced release is probably too low to detect by ELISA (only mRNA levels are shown in Fig. 4E and show a much lower induction compared to Fig. 4A). As all the behavioral assays are performed with LPS injection, the authors need to compare the effects of LPS on cytokine release from WT and KO astrocytes (see above). Importantly, how does acute inhibition of Orai1, i.e. with BTP-2 affect LPS induced release from WT or KO astrocytes?

6. Are the similar results on release of inflammatory cytokines from astrocytes observed when culturing reactive astrocytes without using the AWESOM protocol? LPS treatment in WT mice induces reactive astrocytes as shown in Fig. 5, is it possible that KO of Orai1 blocks the transition into reactive astrocytes also in cultures?

7. Fig.2 F-H: absent NFAT translocation is expected in the absence of Orai1, novelty?

8. Do mice with Orai1 deleted microglia show differences in LPS induced depression like behavior and vice versa, do Orai1 deficient astrocyte mice show altered neuropathic pain?

9. Since there are significant gender-specific differences seen in neuropathic pain (Tsujikawa et al., 2023) and in inflammation induced depression by LPS (this MS), from which gender was the RNA-Seq data derived? Are there gender specific differences in the RNAseq data isolated from male or female mice? Can gender-specific differences be seen in cytokine release of cultured astrocytes, potentially treated with BDNF?

10. Figure S1 of the current manuscript is a repeat of data as shown in Toth et al; Fig. 1D indeed is nearly identical to Fig. 1D in Toth et al., and the Thrombin response (Fig. 1I) a repeat of Fig. 2C in Toth et al. Supplementary Figure 7A-D a repeat of recordings done in Toth et al. While it is excellent that data can be reproduced, also in a different mouse model, all data with GFAP-Cre should be moved to the supplemental figures.

Reviewer #2 (Remarks to the Author):

In this work Murali Prakriya et al. identify Orai1 calcium channels regulate astrocyte metabolic reprogramming and pathogenic function in the context of neuroinflammation and inflammation-induced depression.

Authors represent astrocytes lacking Orai1 shows significant decrease of pro-inflammatory signals (IL-1 α , TNF- α , IL-6, MCP-1, and MCP-1 α) and further induces metabolic reprogramming. In line with these observations, astrocyte Orai1 KO mice reduces CNS inflammation in LPS-evoked neuroinflammation model. Furthermore, Astrocyte Orai1 channels have been shown to regulate inflammation-induced depression behaviors.

This manuscript should be of interest to the broad readership of Nat communications.

Reviewer #3 (Remarks to the Author):

Regulation of astrocyte-mediated brain inflammation by Orai1 channels.

Novakovic et al.

Astrocytes are a major CNS cell type and are increasingly recognized as important components of the CNS response to injury and disease, adopting what is commonly referred to as a 'reactive' phenotype. However, the molecular mechanisms underlying reactive astrogliosis are largely unknown. Ca²⁺ release from internal stores, in an IP₃ dependent manner, is thought to be central to astrocyte function. Following depletion of internal Ca²⁺ stores, ongoing signaling must be maintained by store-operated Ca²⁺ entry (SOCE). Navakovic and

colleagues have investigated the role of SOCE in regulating astrocyte reactivity by genetically ablating a key subunit of the Ca²⁺ release activated Ca²⁺ channel (CRAC) system, Orai1. Novakovic and colleagues report that deletion of Orai1 from astrocytes downregulates the expression of key glycolytic enzymes, metabolic intermediates and impairs ATP production. Furthermore, in their hands, Orai1 deletion reduces cytokine production in the hippocampus, reduces reactive astrogliosis and impacts inhibitory synaptic transmission in hippocampal CA1. Building on these observations, Novakovic et al., then report that mice with astrocyte-specific Ora1a are protected against inflammation induced depression.

Given the increasing recognition of astrocytes as key players in CNS disease, the manuscript is timely and the central theme of the manuscript appears novel. The manuscript itself contains a large volume of work, which has obviously required a large investment of time and effort on behalf of the authors. However, in my opinion, there are issues which need to be addressed before the manuscript can be considered for publication in Nature Communications.

Major issues:

(i) For the central claims of the manuscript to be valid, Orai1 deletion must be cell autonomous and exclude potential off-target effects. However, in my opinion, the authors provide no such evidence and we are left to believe the specificity of the Aldh1l1-CreER^{T2} and GFAP-Cre lines. While these lines have been widely used, it is my opinion, that the authors should still demonstrate specificity in their hands. Otherwise, isn't it suspicious that expression of IBA1 (a Ca²⁺ binding protein) is reduced in Orai1 cKO mice compared to wild types? Or is this just coincidental? Along similar lines, I do not see any indication of how pure their astrocyte cultures are. Surely, it is important to exclude microglial contamination when measuring cytokine levels (either RNA or protein)? Finally, and a more nuanced point, even if Cre-mediated recombination is/was limited to astrocytes, the time course of protein turnover and how this relates to the experimental paradigms is not fully evident (as it was only assessed in cultured cells: see below). Perhaps, the authors could also comment on why tamoxifen was added to cultures: I thought the active metabolite 4-hydroxy tamoxifen needed to be added. How does this impact their results?

(ii) The authors make extensive use of cultured astrocytes in their work. Whether cultured astrocytes fully recapitulate *in vivo* astrocytes is hotly debated (for example, see Foo et al., Neuron, 2011). In an attempt to offset this criticism, the authors use "AWESAM" astrocytes which they claim are "stellate astrocytes with complex morphology, long processes and a more *in vivo* like transcriptome". The original paper describing "AWESAM" astrocytes showed that these cells express high levels of proteins associated with vesicle trafficking (Wolfes et al., J Gen Physiol, 2017), which is not the situation for *in vivo* hippocampal astrocytes (Chai et al., Neuron, 2017). Furthermore, the author's own data showing cell morphology (Fig. 4G) does not correspond to a "stellate" structure. Coupled with the extreme treatments used (e.g. prolonged PDBu exposure) the authors should, in my opinion, be much more circumspect with the conclusions they draw. While this could be offset by appropriate *in vivo* measurements, this type of experiment is generally lacking.

(iii) Personally, I am not sure if the data presented really offer a mechanistic explanation for what is observed. The authors refer several times to Ora1 being a “key checkpoint” for pro-inflammatory cytokine production. However, this protein is involved in refilling ER Ca²⁺ stores, as emphasized by the authors, so what exactly is the link to increased transcription and, perhaps more importantly, cytokine release? In this respect, the paper feels slightly superficial.

(iv) The lack of observed depressive effects induced by inflammation in female wild type mice impacts the global significance of the study, and should be more thoroughly addressed by the authors.

Minor issues:

(i) In general, the manuscript would benefit from tidying. There were several instances of incomplete text (e.g. pSIRV-NFAT-eGFP and was a gift: page 35), figures were cited out of order in the text and some were missing (Fig. 6J: page 17), references were incorrectly cited (there is no Ref 79 listed: Supp Figure 2). Are mouse genotypes really correctly cited with appropriate nomenclature?

(ii) In general, Ca²⁺ signaling in astrocytes is much more complex than presented by the authors in the ‘Introduction’ – see for example, the various types of Ca²⁺ measured in hippocampal and striatal astrocytes (see Chai et al., Neuron, 2017).

(iii) My personal opinion is that some of the authors claims are not substantiated by the data. mRNA levels do not reflect protein levels, and deletion of one channel subunit could affect the stability of other subunits at the protein level (Page 6). Likewise, does the *in vitro* calibration for fura-2 accurately reflect the *in vivo* situation, or is this an approximation? (see Helmchen, CSH Protocols, 2011).

(iv) Images of GFAP, IBA1 and C3 levels in hippocampal slices are not convincing. Would larger images work better? Even assuming that the GFAP and IBA1 responses were supporting extensive gliosis, why is the C3 signal so low (Figure 5 and Supp Fig 6).

(v) The electrophysiological measurements show an interesting effect on excitatory and inhibitory transmission but the measurements appear superficial. Why was the analysis limited to mini-analysis?

(vi) Could the authors speculate in the “Discussion” about the potential therapeutic aspects to their work?

Reviewer 1.

We thank the reviewer for their constructive comments which we believe have significantly improved the manuscript.

1. Is the reduced ATP release seen in Orai1 deleted astrocytes (Toth et al) due to reduced release (Toth et al) or due to an overall increase in AMP/ATP ratios and reduced ATP (Fig 3 F,G, current MS) and thereby leading to a reduced vesicular ATP concentration? Same applies for the reduced glutamate release reported in Toth et al. which may be due to altered glutamine metabolism. In Toth et al., a direct role of SOCE in release of vesicles was postulated. Is this result still valid?

In Toth et al (Toth et al., 2019), we directly monitored vesicle exocytosis using synaptophlourin (spH) (a modified GFP that is directly tethered to the intracellular face of the vesicular protein, VAMP). With this tool, we found that stimulating Orai1 channels in astrocytes with GPCR agonists or by direct store depletion strongly triggers vesicular exocytosis and this is lost in Orai1 cKO astrocytes. Therefore, yes, we do believe that Orai1-mediated SOCE has a direct role in stimulating vesicular exocytosis including the release of ATP through vesicular fusion. But as the reviewer correctly points out, in the current manuscript, we also find that ATP synthesis itself is reduced following ablation of Orai1 in stimulated astrocytes. Thus, we believe that the reduction in stimulus-evoked ATP release that we found in Toth et al in Orai1 KO astrocytes is due to both suppression of ATP production, and the inhibition of its release. In addition to vesicular exocytosis, ATP release has many mechanisms including release through pannexin channels that does not rely on vesicular exocytosis. Thus, multiple mechanisms may underlie the original observation (Toth et al., 2019) that Orai1 cKO astrocytes show reduced stimulus-evoked ATP release. We have added a brief discussion of this point in the Discussion which we hope clarifies the issue.

2. Inhibition of GABA interneurons (as seen by Toth et al) has been shown to be sufficient and necessary for an antidepressant response (Fogaca et al., Mol Psych. 2021). Is this the explanation for the behavioral phenotype of the Orai KO astrocytes, potentially independent from inflammatory aspects? What happens if stress stimuli and not inflammation are used to trigger depression?

This is an excellent point, and we are glad that the reviewer brings this up. In a nutshell, no we do not think that simply the loss of interneuron stimulation in Orai1 cKO mice by astrocyte stimulation can fully explain the behavioral phenotype. Here, in the LPS inflammation model, we find that the activity of inhibitory synaptic activity significantly decreases *below* that baseline levels in saline treated animals (Figure 7G). Thus, not only is stimulation of inhibitory interneurons by thrombin-activated astrocytes lost in Orai1 cKO mice (Toth et al., 2019), but also in response to LPS, the astrocyte Orai1 cKO mice show strong decreases in IPSC activity well below levels in the saline exposed cKO mice. There is also a substantial decline in the amplitude of the sIPSCs, which was not seen when astrocytes were stimulated with thrombin in non-LPS treated mice (Toth et al., 2019). Thus, the antidepressant phenotype of the Orai1 cKO mice is not simply due to lack of thrombin-mediated and astrocyte-evoked sIPSC *facilitation* that we previously described (although that could also be a smaller contributing factor). Instead, the robust *decrease* in sIPSC activity in the cKO mice following LPS administration is what likely drives the antidepressant response, in line with the findings of Fogaca et al (Fogaca et al., 2021). We attribute this aspect of

the synaptic change to the unmasking of inhibition by cytokines such as IL1 β which are produced by macrophages, and which would be expected to be still operational in astrocyte Orai1 KO mice.

3. Release of many pro-inflammatory cytokines usually entails activation of P2RX7 channels and indeed, acute stress induces depressive-like behavior has been shown to be mediated by astrocytic and microglial P2X7 channels (Zhao YF et al., 2022, see also 51 citations on astrocytes, P2X7 and inflammatory cytokines). These data need to be discussed and using the available RNA-Seq data, the authors need to show in immunohistochemical staining of adult astrocytes from tamoxifen induced Orai1KO (or GFAP-Cre) if levels of P2X7, and some selected components such as pannexins, connexins, TLR receptors (or other pathways of inflammation (Fig. 2B)) are altered in vivo. If downregulated, mediators of the classic LPS triggered NLRP3 inflammasome, potentially indirectly protect mice from inflammation induced anhedonia, helplessness and despair. In spinal slices, astrocytic connexin43 hemichannels respond to LPS (Panattoni et al., Mol Brain 2021).

Again, an excellent point and we thank the reviewer for bringing this up. To address this point, we carried out careful analysis of the RNA-seq results to assess whether the ablation of Orai1 changes expression of TLR4, pannexins, P2RX4, P2RX7, NLRP3, and other molecules implicated in the inflammasome signaling pathway. As shown in the adjoining Figure 1A, we did not detect changes in the expression of these or related proteins in the cKO astrocytes compared to WT

Figure 1: RNA-Seq analysis of select molecules involved in the LPS inflammatory response. (A) Expression of key genes involved in LPS induced inflammatory responses and inflammasome pathway (TLR4, Nlrp3, P2RX7, P2RX4.) Expression was not altered in Orai1 cKO astrocytes. **(B)** By contrast, stimulated astrocytes showed marked downregulation of several genes including P2RX7, TLR4, and Panx2 in both WT and Orai1 cKO astrocytes.

astrocytes. This result indicates that the strong reduction in LPS-induced cytokine production that we see in the Orai1 cKO mice is not due to changes in expression of the LPS involved molecules. Interestingly, however, cell stimulation with thapsigargin and PdBu downregulated several molecules including Panx2, P2RX7, and TLR3 in both WT and KO astrocytes (Figure 1B, above). Thus, while expression of proteins in the LPS signaling pathway is dependent on the activation state of the cell, this effect is not Orai1 dependent. We conclude from these results that molecules involved in the LPS/inflammasome pathway are not regulated by Orai1 channels.

In another set of experiments, we carried out immunohistochemistry (IHC) of brain hippocampal brain slices for key molecules in the TLR4-P2X7 pathway including P2RX7, TLR4,

and connexin 43 (Figure 2). The IHC showed diffuse staining for these molecules mostly in the neuropil of the CA1 region but also in the principal neuronal layer for P2RX7 and TLR4. We did not notice any obvious difference in staining in slices from WT vs Orai1 cKO mice. Altogether, we conclude from this analysis that the reduced response to LPS in the Orai1 cKO mice is not simply due to downregulation of these proteins. The results rather suggest that Orai1 signaling synergizes with the signaling mediated by TLR4 to drive cytokine synthesis.

Figure 2: Immunohistochemistry of P2X7, TLR4, and Connexin 43 in the hippocampus in . Immunostaining of hippocampal brain slices for P2RX7, TLR4, and Connexin43. We saw broad labelling of these proteins in the CA1 region of the hippocampus and this was not affected by ablation of Orai1. Coronal cut brain slices (30 μ m thick) were labelled with antibodies for the indicated proteins.

4. How do levels of released cytokines compare in WT and KO cells when using a classical dual activation protocol (i.e. LPS+ATP)?

To address this question, we carried out experiments using ELISA to measure cytokines released from primary astrocyte cultures after they were stimulated with other LPS, LPS+ATP, or the proinflammatory GPCR agonist, thrombin (Figure 3, below). We found that LPS substantially induced the release of several cytokines from astrocytes including IL-6, MCP1, MIP-1 α , and IL-1 α . In all cases, we did not find additional effects of stimulating cells with ATP (on the top of LPS) compared to LPS alone indicating that at least under our conditions, a dual activation protocol with ATP and LPS does not favor inflammasome activation. This result suggests that in contrast to macrophages or microglia, astrocytes are not professional inflammasome-regulated cells. Most importantly, deletion of Orai1 significantly reduced induction of IL-6, IL1 α , and MIP1 α but not of the other mediators that we examined. Thus, under conditions where LPS directly stimulates astrocytes, Orai1 makes only small but likely physiologically important contributions to

Figure 3: LPS-mediated induction of proinflammatory cytokines is reduced in Orai1 cKO astrocytes. Primary astrocytes (WT or Orai1 KO) were stimulated with either LPS, LPS+ATP or Thrombin and cytokine levels were measured in the supernatant via ELISA. Ablation of Orai1 suppresses release of many cytokines induced by LPS and thrombin.

inflammatory cytokine production. **We should note that these effects are not likely relevant for the whole-animal peripheral LPS injection studies we carried out in Figures 5, 6, 7, and 8 of the manuscript because LPS is not known to cross the blood brain barrier (BBB).** Instead, the observed LPS-induced effects on brain inflammation are likely driven by peripheral cytokines and mediators (including thrombin) that readily cross the BBB to induce astrocyte activation.

5. Overall, the amounts of released cytokines are low (~50 pg/ml for IL-6 after Tg/PDBu, Fig. 4B, compared to ~3200 pg/ml after LPS (Lu, X et al, J Neuroinflamm.) or 400 pg/ml after Il-1 α /TNF α stimulation (Nakajima et al., 2022), in this later report, release of PGE2 was shown to be enhanced by KD of Orai2. In addition, Thrombin induced release is probably too low to detect by ELISA (only mRNA levels are shown in Fig. 4E and show a much lower induction compared to Fig. 4A). As all the behavioral assays are performed with LPS injection, the authors need to compare the effects of LPS on cytokine release from WT and KO astrocytes (see above). Importantly, how does acute inhibition of Orai1, i.e. with BTP-2 affect LPS induced release from WT or KO astrocytes?

We fully agree. In the new experiment shown above done in parallel on the same batch of cell, LPS induced 2-fold greater release of IL-6 than that caused by thrombin (~1000 pg/ml vs 500 pg/ml) which are in turn greater than the levels induced by Tg/PdBu. However, the finding that deletion or blockade of Orai1 impairs the induction of cytokines remains constant across the stimulation conditions, whether through LPS, thrombin, or Tg/PdBu. Both LPS and thrombin obviously induce signaling pathways distinct from and in addition to Orai1 channel activation by Ca²⁺ store depletion, but the results suggest that these signals synergize for effective cytokine synthesis such that in the absence of Orai1-mediated SOCE, LPS and thrombin-evoked cytokine synthesis is reduced.

To assess how acute inhibition of Orai1 by BTP2 affects LPS induced release of cytokines, we pretreated cells with BTP2 and then added LPS (or LPS+ATP) in parallel with the other experiments shown in Figure 3 above. We found that BTP2 strongly suppresses IL-6, MCP1, MIP1 α production (more or less mirroring effects seen with ablation of Orai1). Together, these results indicate that Orai1 makes critical contributions to cytokine production in response to LPS stimulation.

6. Are the similar results on release of inflammatory cytokines from astrocytes observed when culturing reactive astrocytes without using the AWESOM protocol? LPS treatment in WT mice induces reactive astrocytes as shown in Fig. 5, is it possible that KO of Orai1 blocks the transition into reactive astrocytes also in cultures?

To address this point, we cultured astrocytes using the old-school method in the presence of serum (FBS, Figure 4). These astrocytes show a large footprint lacking the stellate shape long processes characteristic of endogenous astrocytes. Like astrocytes cultured with the AWESAM method, we found that depletion of ER Ca²⁺ stores to evoke Orai1 activation in these serum-grown cells caused the production of IL-6, MIP-1 α , and TNF α , similar to the effects seen in AWESAM astrocytes. But the extent of upregulation of cytokines following cell stimulation was smaller. We attribute to

Figure 4: Cytokine analysis of astrocytes grown in the presence of FBS. (A) Morphology of astrocyte grown in the absence (AWESAM) or presence of serum (FBS). Cells were stained for GFAP with a monoclonal Ab. **(B)** Cytokine induction in the astrocytes grown in the presence of serum (FBS). Thapsigargin/PdBu treatment stimulates induction of IL-6, MIP-1 α , and TNF- α .

this difference to the possibility that in the presence of serum, astrocytes are likely already in a stimulated state where further stimulation evokes a smaller response. We prefer to use the astrocytes grown in the AWESAM protocol because the cells are morphologically closer to astrocytes with stellate processes and the analysis by the Dean group has previously indicated that their transcriptome is more similar to that of in vivo astrocytes than the astrocytes grown continuously in the presence of serum (Wolfes et al., 2017; Wolfes and Dean, 2018).

Finally, is it possible that the KO of Orai1 blocks transition of astrocytes into the reactive state? Yes, definitely. We assessed this question through the expression of so-called pro-resolving “A2” and pro-inflammatory (or reactive) “A1” markers in the WT and Orai1 cKO cells (Liddelw and Barres, 2017; Liddelw et al., 2017). We found that deletion of Orai1 preferentially reduced expression of several A1, but not A2 markers (Figure 5). Although the classification of astrocytes into A1 or A2 is controversial and has fallen out of favor recently (Escartin et al., 2021), these results do suggest that signaling by Orai1 drives astrocytes towards the inflammatory A1 state.

		A1 reactive markers											
		Amg2	Fab5	Fabp5	Gpr2	Orai1	H2-D1	H2-T23	Igfbp1	Pendb	Prkn	Scarb1	Ucp1
WT		1.8±0.4	3.0±0.3	1.1±0.1	2.2±0.7	0.2±0.2	2.6±0.2	3.4±0.5	0.5±0.1	0.4±0.1	0.6±0.0	0.3±0.2	0.8±0.2
Orai1 KO		0.2±0.2	2.3±0.2	0.8±0.1	0.8±0.3	1.4±0.3	1.5±0.2	1.2±0.3	1.0±0.3	1.7±0.3	2.3±0.2	1.6±0.3	2.0±0.2
		A2 reactive markers											
		B2m1b	Ccl10b	Ccl14	Ccl11	Empr1	Piqa2	Pxa3	S100a10	Ser100b	Sh2b2	Tgm1	Tnfrsf1
WT		0.8±0.4	0.5±0.2	0.4±0.3	2.7±0.2	2.8±0.4	1.9±0.8	1.8±0.6	0.9±0.3	0.1±0.1	0.2±0.3	0.3±0.2	0.4±0.4
Orai1 KO		1.8±0.3	0.0±0.2	0.2±0.1	2.4±0.6	0.2±0.7	2.5±0.8	2.2±0.9	1.6±0.4	0.3±0.2	0.2±0.3	0.3±0.2	0.8±0.3

* = p<0.05 (Mann-Whitney U test)

Figure 5: Comparison of A1 vs A2 markers. Astrocytes were stimulated with TG/PdBu to evoke Orai1 activation and the A1 and A2 markers described by Liddelw were analyzed in WT and Orai1 cKO cells.

7. Fig.2 F-H: absent NFAT translocation is expected in the absence of Orai1, novelty?

Orai1 is strongly linked to NFAT activation in many types of immune cells (Hogan et al., 2003), though it has not been demonstrated in astrocytes to our knowledge. Thus, it is not entirely surprising that Orai1 regulates NFAT activation and NFAT-dependent gene expression in astrocytes. The novelty of our finding is that we have linked NFAT activation to a specific Ca²⁺ influx pathway (Orai1) in astrocytes. Moreover, because NFAT activation in astrocytes is linked to exaggerated endpoints in several disease models including Alzheimer’s Disease and traumatic brain injury (TBI) (Sompol and Norris, 2018), we speculate that the suppression of Orai1 signaling may mitigate some of the negative effects of NFAT activation in these and disease states. This finding represented only a small aspect of our paper and hence we did not dwell on it.

8. Do mice with Orai1 deleted microglia show differences in LPS induced depression like behavior and vice versa, do Orai1 deficient astrocyte mice show altered neuropathic pain?

Excellent point. We have not yet investigated the effects of Orai1 deletion in microglia on LPS induced depression, but we expect that there might be a similar protective effect as that seen in the astrocyte Orai1 KO mice. We intend to carry out these studies which will be addressed separately. We have also have not addressed whether Orai1 deficient mice show altered neuropathic pain which will be determined out in an ongoing, separate study.

9. Since there are significant gender-specific differences seen in neuropathic pain (Tsujikawa et al., 2023) and in inflammation induced depression by LPS (this MS), from which gender was the RNA-Seq data derived? Are there gender specific differences in the RNAseq data isolated from male or female mice? Can gender-specific differences be seen in cytokine release of cultured astrocytes, potentially treated with BDNF?

Figure 6. Comparison of cytokine induction in astrocytes from male and female mice. Astrocytes were separately cultured from individual mice as described in the paper. Regardless of the sex of the mouse, cell stimulation with Tg/PdBu caused strong induction of numerous cytokine genes which was suppressed to similar extents in male and female astrocytes. This result indicates that astrocyte Orai1 channels make equal contributions to inflammatory cytokine synthesis in both sexes.

Again, an excellent point and we thank the reviewer for bringing this up. To analyze this question, we examined the RNA-seq results. We found that there were no obvious gender-dependent differences between cells obtained from male mice vs female mice in the heat maps (please see Figure 2C and Supplementary Figure 4H of manuscript). Furthermore, PCA analysis of the RNA-seq samples show overlap of male or female mice, indicating that sex is not driving differential gene expression *in vitro* (Supplementary Figure 4C). Although our sample size is too small to definitively say so with certainty, these heat maps do not reveal any obvious gender differences. Moreover, a direct examination of cytokine induction between astrocytes from male and female mice revealed that in both sexes, astrocyte stimulation with thapsigargin and PdBu stimulates the induction of IL-1 α , IL-6, IL-33, MCP1, MIP-1 α , and TNF α (Figure 6 above). In each case, the deletion of Orai1 suppresses induction of these cytokines in both male and female mice. Thus, at least at the level of this *in vitro* analysis, we do not find evidence for sexual dimorphism. Moreover, *in vivo* also, we found that deletion of Orai1 affected the upregulation of GFAP and IBA1 equally in both male and female mice (please see Figure 5 and Supplementary Figure 8 of the manuscript). These results indicate that Orai1 makes similar contributions to astrocyte reactivity in male and female mice.

10. Figure S1 of the current manuscript is a repeat of data as shown in Toth et al; Fig. 1D indeed is nearly identical to Fig. 1D in Toth et al., and the Thrombin response (Fig. 11) a repeat of Fig. 2C in Toth et al. Supplementary Figure 7A-D a repeat of recordings done in Toth et al. While it is excellent that data can be reproduced, also in a different mouse model, all data with GFAP-Cre should be moved to the supplemental figures.

Done.

Reviewer 3.

We thank the reviewer for thoughtful comments and recommendations.

1. For the central claims of the manuscript to be valid, *Orai1* deletion must be cell autonomous and exclude potential off-target effects. However, in my opinion, the authors provide no such evidence and we are left to believe the specificity of the *Aldh111-CreERT2* and *GFAP-Cre* lines. While these lines have been widely used, it is my opinion, that the authors should still demonstrate specificity in their hands. Otherwise, isn't it suspicious that expression of *IBA1* (a Ca^{2+} binding protein) is reduced in *Orai1* cKO mice compared to wild types? Or is this just coincidental? Along similar lines, I do not see any indication of how pure their astrocyte cultures are. Surely, it is important to exclude microglial contamination when measuring cytokine levels (either RNA or protein)? Finally, and a more nuanced point, even if Cre-mediated recombination is/was limited to astrocytes, the time course of protein turnover and how this relates to the experimental paradigms is not fully evident (as it was only assessed in cultured cells: see below). Perhaps, the authors could also comment on why tamoxifen was added to cultures: I thought the active metabolite 4-hydroxy tamoxifen needed to be added. How does this impact their results?

We thank the reviewer for bringing up these important controls. Since several individual points were raised above, we address them one-by-one below:

(i) Off target effects of the Cre-mediated deletion of *Orai1* by *Aldh1-Cre* and purity of the astrocyte cultures. *Aldh111-Cre* is known to be a highly selective delete strain target astrocytes, and a previous study has explicitly shown that using *Aldh111-CreERT2*, no Cre mediated deletion was detected in microglia (Winchenbach et al., 2016). However, to directly address in our experimental system the possibility that *Aldh111-CreERT2* may affect *Orai1* expression in microglia, we explicitly carried out a test to evaluate microglial *Orai1* deletion. We delivered 4OH-tamoxifen to microglia from *Orai1^{fl/fl} Aldh111-CreERT2* mice and assessed *Orai1* expression in microglia using qPCR. As shown in Figure 7A below, no decrease in *Orai1* was detected by qPCR. This is in contrast to strong suppression of *Orai1* mRNA with the same manipulation in astrocytes (see Figure 1 of paper). We note that the *Aldh111 Cre* based approach also did not affect expression of the other SOCE components including *Orai2*, *Orai3*, *STIM1*, or *STIM2* in astrocytes. Based on both this analysis and previous reports, we are confident that the targeted ablation of *Orai1* in astrocytes does not directly affect microglia or the other SOCE molecules in astrocytes.

So how does *Orai1* deletion in astrocytes affect microglial activation? Ablation of *Orai1* in astrocytes significantly reduced expression of complement factor C3 (Supplementary Figure 9). Because C3 is a potent activator of microglia, a reduction of astrocyte-derived C3 is predicted to also reduce microglial

Figure 7: Specificity of *Orai1* deletion. (A) To assess whether the *Orai1^{fl/fl} Aldh111-Cre* line affects *Orai1* expression in microglia, we cultured microglia from *Orai1^{fl/fl} Aldh111-Cre* mice treated with tamoxifen and examined *Orai1* expression by real-time PCR. *Orai1* expression is unaffected in microglia in this line. (B) Purity of astrocytes. We stained our primary astrocyte cultures for GFAP (astrocyte marker) and IBA1 (microglial marker). No IBA1 staining was visible in our cultures. The right panel shows microglia isolated from the same mouse stained for IBA1.

activation. This finding is consistent with growing evidence that astrocytes and microglia interact in many ways to amplify cascades of inflammation (Jha et al., 2019). We believe this could be one key reason for why we see reduced IBA1 labelling (indicative of reduced microgliosis) in the *Orai1* cKO mice following LPS challenge.

Finally, we note that for inducing *Orai1* deletion in cultured astrocytes, we do use 4-OH tamoxifen (as indicated in the Methods, Catalog# H7904, Sigma-Aldrich, USA) that is added to the culture medium. We inadvertently referred to this as tamoxifen in the text and we apologize for this oversight. We have corrected this in the main text.

(ii) Purity of astrocyte cultures. We assessed this by staining our astrocyte cultures for the microglial marker, IBA1. As shown above in Figure 7B above, in our typical astrocyte culture, we consistently observed that there is little or no IBA1 staining indicating that the culturing method, which involves vigorous shaking to remove the microglia from the astrocyte cell layer during the first week of culture, is highly effective in yielding a relatively pure astrocyte cultures.

(iii) Time course of protein turnover following Cre mediated deletion. This is a good point. To determine whether the *Aldh1l1*-CreERT2 mediated deletion of *Orai1* stably reduces *Orai1* protein expression in astrocytes, we carried out immunohistochemistry for *Orai1* in hippocampal slices. In the WT animals, as expected, the most significant area of *Orai1* expression was found in the CA1 neuronal layer. To analyze the effects on astrocyte *Orai1* expression, we analyzed the *stratum radiatum* and *stratum oriens* layers where astrocytes are enriched in the hippocampus, and quantified *Orai1* expression based on co-localization of the *Orai1* signal with the GFAP signal (which labels an intermediate filament protein) (Figure 8 below). This analysis clearly revealed that *Orai1* expression two weeks after tamoxifen injection was strongly diminished by >80%

Figure 8: Analysis of *Orai1* protein expression by IHC. (A) *Orai1* is expressed in the CA1 hippocampus, particularly near the pyramidal layer (PL), but also in the *stratum oriens* (SO) and *stratum radiatum* (SR). (B) Higher magnification image shows astrocytes in the SR region co-labeled with *Orai1* in WT mice. cKO mice have less *Orai1*-gfap colocalization than WT mice. Scale bar = 25 μ m. (C) Cells co-labeled with GFAP and *Orai1* were quantified in the SO and SR layers using a 5 μ m thick stacked image. Nuclei of GFAP+ cells were identified and cells with *Orai1* staining within 5 μ m of the nuclei were considered *Orai1*+ astrocytes. (Data are mean \pm SEM. n=4 mice/group; >100 cells/mouse. p=8.9x10⁻⁴).

indicating that the *Aldh1l1*-CreERT2 mediated deletion of *Orai1* stably reduces protein expression. These new results are shown in Supplementary Figure 3.

2) *The authors make extensive use of cultured astrocytes in their work. Whether cultured astrocytes fully recapitulate in vivo astrocytes is hotly debated (for example, see Foo et al., Neuron, 2011). In an attempt to offset this criticism, the authors use “AWESAM” astrocytes which they claim are “stellate astrocytes with complex morphology, long processes and a more in vivo like transcriptome”. The original paper describing “AWESAM” astrocytes showed that these cells*

express high levels of proteins associated with vesicle trafficking (Wolfes et al., *J Gen Physiol*, 2017), which is not the situation for *in vivo* hippocampal astrocytes (Chai et al., *Neuron*, 2017). Furthermore, the author's own data showing cell morphology (Fig. 4G) does not correspond to a "stellate" structure. Coupled with the extreme treatments used (e.g. prolonged PDBu exposure) the authors should, in my opinion, be more circumspect with the conclusions they draw. While this could be offset by appropriate *in vivo* measurements, this type of experiment is generally lacking.

We agree that the use of astrocyte cultures in some of the paper's early experiments does raise questions since astrocytes in culture are thought to be significantly different from *in vivo* astrocytes. We do think that the AWESAM astrocyte provide a reasonable approach for understanding some mechanistic questions related to Orail contributions for metabolism, gene expression, and NFAT activation. which was the focus of the paper's mechanistic experiments in the first part of the study. However, we want to emphasize that in the second half of the paper (Figures 5-8), we exclusively used *in vivo* and *in situ* contributions of Orail in astrocytes in their native environment. This includes a large amount of behavioral data in awake, behaving mice (Figure 8). These data together converge on the conclusion that Orail drives astrocyte mediated inflammation to promote brain inflammation and alterations in cognitive functions. Furthermore, while we agree that while the AWESAM astrocytes do not exhibit the dramatic, complex branches seen *in vivo*, their appearance is strikingly distinct from traditional, serum-grown astrocytes as seen in Figure 4 of this response document.

To provide an additional level of control for the PdBu effect for the concern noted above, we carried out experiments to examine gene expression evoked by PdBu alone. These experiments showed that in the absence of concomitant Orail activation, PKC activation alone by PdBu failed to activate any of the key cytokines we examined (Figure 9 below) including IL-6, IL1 α , and TNF α (note that these cytokines all have NFAT binding sites on their promoters). This indicates that PKC alone regulates gene expression to a much lower extent than concomitant activation of PKC with Orail activation. Because the experiments in Figure 3 compared the effects of store depletion

Figure 9: PKC activation alone (with PDBu) without concomitant Orail activation fails to induce inflammatory cytokine production. Astrocytes were treated with PDBu for 6 hours and cytokine mRNA levels were assessed by real-time PCR. treatment in WT and Orail KO cells, we believe that any differences are attributable to the presence of Orail in WT cells.

3) Mechanistic connection of Orail mediated Ca²⁺ signaling to inflammation and behavioral changes.

To address this point, we undertook an ambitious set of experiments to examine two questions: (i) how does peripheral inflammation by LPS affect Ca²⁺ signaling in astrocytes? (ii) How are these Ca²⁺ signals *in situ* affected by ablation of Orail? We performed these experiments using 2-photon laser scanning microscopy (2PLSM) and the genetically encoded Ca²⁺ indicator, GCaMP6f expressed in mice via an astrocyte selective AAV promoter in young adult mice. This powerful approach preserves the structural and functional morphology of astrocytes in the native environment and provides a more realistic picture of how inflammation affects astrocytes and the role of astrocyte Orail channels in this process.

These new data are shown in Figure 6 of the paper and Figure 10 below. The data provide several new insights: 1) LPS caused a significant enhancement of astrocyte Ca^{2+} signaling, which is apparent in the frequency of Ca^{2+} fluctuations in the soma, primary branches, and distal branches. 2) These LPS-evoked increases in astrocyte Ca^{2+} signaling activity were nearly completely blocked in the Orai1 cKO astrocytes. 3) Baseline Ca^{2+} fluctuations were not affected by the deletion of Orai1, but only the challenge (LPS)-evoked Ca^{2+} activity was affected. Because Ca^{2+} is the primary substrate for astrocyte signaling, these results show that peripheral LPS mediated inflammation strongly enhances CNS astrocyte signaling and activation through increased activation of Orai1 calcium channels. We believe that the blockade of astrocyte Ca^{2+} signaling increases by LPS in the cKO mice fill an important gap in the paper and better help link the changes in effector functions (cytokine release, inflammation) to Ca^{2+} signaling. We hope that the reviewer will judge these new results as addressing the concern.

Figure 10: Orai1 channels regulate LPS-evoked increases in astrocyte Ca^{2+} signaling. (A) Schematic illustrating experimental protocol. GCaMP6f was expressed in astrocytes of the hippocampal CA1 using stereotaxic injections of AAV5 virus with an astrocyte-specific *gfaAB1D* promoter. After 2 to 3 weeks to allow for expression, mice were injected with either 1 mg/mL LPS or equivalent volume of saline. 24-hours following intraperitoneal LPS injection, Ca^{2+} fluctuations in CA1 astrocytes expressing GCaMP6 were imaged using 2PLSM. (B) Examples of Max-IP images of astrocytes transfected with gCaMP6f in the CA1 region of the hippocampus. (C) Sample traces from the soma, primary branches, and distal branches of astrocyte Ca^{2+} fluctuations in brain slices from WT+saline, WT+LPS, cKO+saline, and cKO+LPS mice. (D) Summary graphs for frequency of calcium oscillations calculated over three minutes of imaging. Each dot represents one ROI (amplitude is the average of all peaks in one ROI, frequency is total # of events/3 minutes). WT saline: n=3 mice, 16 cells; WT+LPS: n=3 mice, 15 cells; cKO+saline: n=3 mice, 11 cells; cKO+LPS: n=3 mice, 13 cells. * = $p < 0.05$, ** = $p < 0.01$, *** = $p < 0.005$.

4) *Lack of depression type behaviors in female mice.* Other labs have seen this behavioral difference (Mello et al., 2018; Millett et al., 2019) and recent papers have uncovered the underlying basis of this differential behavioral depression phenotype between male and female mice in response to LPS (Rossetti et al., 2019). This is believed to be likely due to strong protective effect by BDNF in female rodents (Millett et al., 2019). Accordingly, deletion of one allele (heterozygous BDNF mutant female mice) suffices to reveal depression-like phenotypes in response to LPS (Rossetti et al., 2019).

However, we would like to note that in response to LPS, female mice also show increases in brain inflammation as evidenced by increased expression of GFAP and IBA, and levels of C3. Importantly, deletion of *Orai1* blocks the upregulation of these markers for brain inflammation. Additionally, loss of *Orai1* affects gene expression similarly in both male and female mice in the *in vitro* experiments. Thus, our findings showing that ablation of *Orai1* mitigates inflammatory markers in both sexes strongly suggests that targeting astrocytic *Orai1* may offer a path for mitigating brain inflammation in both sexes.

Minor issues:

(i) In general, the manuscript would benefit from tidying. There were several instances of incomplete text (e.g. pSIRV-NFAT-eGFP and was a gift: page 35), figures were cited out of order in the text and some were missing (Fig. 6J: page 17), references were incorrectly cited (there is no Ref 79 listed: Supp Figure 2). Are mouse genotypes really correctly cited with appropriate nomenclature?

We apologize for these errors. We have carefully gone through the manuscript and have corrected the typos indicated above including in the reference citations in the Supplementary Figure legends (since it was a separate document, the references had to be tracked manually which caused the original error). We have also carefully verified the mouse genotypes showed in all figures and I can reaffirm that these are all correctly reported.

(ii) In general, Ca²⁺ signaling in astrocytes is much more complex than presented by the authors in the 'Introduction' – see for example, the various types of Ca²⁺ measured in hippocampal and striatal astrocytes (see Chai et al., Neuron, 2017).

We agree. And this point is now reinforced by the *in-situ* measurements of astrocyte Ca²⁺ signaling that we have now added (Figure 6 of manuscript). These Ca²⁺ signals, monitored using 2-photon laser scanning microscopy show dynamic fluctuations which are amplified by LPS. Deletion of *Orai1* blocks the upregulation of these Ca²⁺ fluctuations as further elaborated for point 3 above.

(iii) My personal opinion is that some of the authors claims are not substantiated by the data. mRNA levels do not reflect protein levels, and deletion of one channel subunit could affect the stability of other subunits at the protein level (Page 6). Likewise, does the in vitro calibration for fura-2 accurately reflect the in vivo situation, or is this an approximation? (see Helmchen, CSH Protocols, 2011).

The *in vitro* measurements are certainly an approximation since the calibration is carried out *in vitro* and in cultured astrocytes. In slices, there is unfortunately no easy way to put exact numbers

of the Ca²⁺ signals using single wavelength dyes such as GCaMP6f. However, the relative changes in fluctuation frequency and amplitude are generally considered a very useful measure of the activity of cell being studied, and we have used this approach to probe the changes in effects of peripheral LPS induced inflammation on astrocyte activity and the consequences of ablating Orai1.

(iv) Images of GFAP, IBA1 and C3 levels in hippocampal slices are not convincing. Would larger images work better? Even assuming that the GFAP and IBA1 responses were supporting extensive gliosis, why is the C3 signal so low (Figure 5 and Supp Fig 6).

To address this problem, we have enlarged the figures and are hoping that these enlarged pictures provide greater clarity. C3 levels are quite low in the brain in normal physiological circumstances (<https://www.proteinatlas.org/ENSG00000125730-C3/tissue>), and this low signal is compounded by the punctate appearance of C3 even in culture. Despite low expression levels, C3 is a powerful inflammatory mediator in the brain. We do see a measurable and robust *increase* in C3 staining after LPS injection. We now show the C3 images without the DAPI stain, so the signal is more clearly visible in Supplementary Figure 9.

(v) The electrophysiological measurements show an interesting effect on excitatory and inhibitory transmission but the measurements appear superficial. Why was the analysis limited to mini-analysis?

To analyze the effects of altered astrocytes Ca²⁺ signaling and inflammatory cytokine production on neuronal function, we recorded spontaneous excitatory and inhibitory synaptic currents using the standard methods described in many of our previous studies and in the literature (Toth et al., 2019; Hori et al., 2020; Maneshi et al., 2020; Fogaca et al., 2021). These are shown in Figure 7 and Supplementary Figure 10, respectively. Since we are not deleting Orai1 in astrocytes but not in neurons, the neuronal genotype should be unaffected and hence we thought that any changes in synaptic activity arise from external influences (rather than cell-autonomous changes). Hence, we decided that it would be best to analyze **spontaneous** synaptic currents that should be influenced by cell extrinsic mechanisms (*not* miniature synaptic currents that are reflective of intrinsic mechanisms). The measurements of excitatory and inhibitory currents actually do provide a fairly extensive description of the effects of astrocyte Orai1 deletion on neuronal synaptic function. We note that in our previous work (Toth et al, 2019), we did look at miniature currents and saw no difference in the Orai1 astrocyte KO slices as would be expected since in the presence of TTX, the residual synaptic currents would only reflect quantal release that occurs at random. Thus, these methods are appropriate for the question being addressed.

(vi) Could the authors speculate in the “Discussion” about the potential therapeutic aspects to their work?

Thank you for the suggestion. We have added a discussion on the potential implications of the findings for human depression and its potential mechanism involving inhibition of inhibitory synaptic transmission, analogous to the effects of rapidly acting antidepressants.

References

- Escartin, C., E. Galea, A. Lakatos, J.P. O'Callaghan, G.C. Petzold, A. Serrano-Pozo, . . . A. Verkhratsky. 2021. Reactive astrocyte nomenclature, definitions, and future directions. *Nat Neurosci.* 24:312-325.10.1038/s41593-020-00783-4
- Fogaca, M.V., M. Wu, C. Li, X.Y. Li, M.R. Picciotto, and R.S. Duman. 2021. Inhibition of GABA interneurons in the mPFC is sufficient and necessary for rapid antidepressant responses. *Mol Psychiatry.* 26:3277-3291.10.1038/s41380-020-00916-y
- Hogan, P.G., L. Chen, J. Nardone, and A. Rao. 2003. Transcriptional regulation by calcium, calcineurin, and NFAT. *Genes Dev.* 17:2205-2232
- Hori, K., S. Tsujikawa, M.M. Novakovic, M. Yamashita, and M. Prakriya. 2020. Regulation of chemoconvulsant-induced seizures by store-operated Orail channels. *J Physiol.* 598:5391-5409.10.1113/JP280119
- Jha, M.K., M. Jo, J.H. Kim, and K. Suk. 2019. Microglia-Astrocyte Crosstalk: An Intimate Molecular Conversation. *The Neuroscientist : a review journal bringing neurobiology, neurology and psychiatry.* 25:227-240.10.1177/1073858418783959
- Liddelow, S.A., and B.A. Barres. 2017. Reactive Astrocytes: Production, Function, and Therapeutic Potential. *Immunity.* 46:957-967.10.1016/j.immuni.2017.06.006
- Liddelow, S.A., K.A. Guttenplan, L.E. Clarke, F.C. Bennett, C.J. Bohlen, L. Schirmer, . . . B.A. Barres. 2017. Neurotoxic reactive astrocytes are induced by activated microglia. *Nature.* 541:481-487.10.1038/nature21029
- Maneshi, M.M., A.B. Toth, T. Ishii, K. Hori, S. Tsujikawa, A.K. Shum, . . . M. Prakriya. 2020. Orail Channels Are Essential for Amplification of Glutamate-Evoked Ca(2+) Signals in Dendritic Spines to Regulate Working and Associative Memory. *Cell reports.* 33:108464.10.1016/j.celrep.2020.108464
- Mello, B.S.F., A.J.M. Chaves Filho, C.S. Custodio, R.C. Cordeiro, F. Miyajima, F.C.F. de Sousa, . . . D. Macedo. 2018. Sex influences in behavior and brain inflammatory and oxidative alterations in mice submitted to lipopolysaccharide-induced inflammatory model of depression. *Journal of neuroimmunology.* 320:133-142.10.1016/j.jneuroim.2018.04.009
- Millett, C.E., B.E. Phillips, and E.F.H. Saunders. 2019. The Sex-specific Effects of LPS on Depressive-like Behavior and Oxidative Stress in the Hippocampus of the Mouse. *Neuroscience.* 399:77-88.10.1016/j.neuroscience.2018.12.008
- Rossetti, A.C., M.S. Paladini, A. Trepci, A. Mallien, M.A. Riva, P. Gass, and R. Molteni. 2019. Differential Neuroinflammatory Response in Male and Female Mice: A Role for BDNF. *Frontiers in molecular neuroscience.* 12:166.10.3389/fnmol.2019.00166
- Sompol, P., and C.M. Norris. 2018. Ca(2+), Astrocyte Activation and Calcineurin/NFAT Signaling in Age-Related Neurodegenerative Diseases. *Front Aging Neurosci.* 10:199.10.3389/fnagi.2018.00199
- Toth, A.B., K. Hori, M.M. Novakovic, N.G. Bernstein, L. Lambot, and M. Prakriya. 2019. CRAC channels regulate astrocyte Ca²⁺ signaling and gliotransmitter release to modulate hippocampal GABAergic transmission. *Sci Signal.* 12 eaaw5450:1-15
- Winchenbach, J., T. Duking, S.A. Berghoff, S.K. Stumpf, S. Hulsmann, K.A. Nave, and G. Saher. 2016. Inducible targeting of CNS astrocytes in Aldh11-CreERT2 BAC transgenic mice. *F1000Res.* 5:2934.10.12688/f1000research.10509.1

- Wolfes, A.C., S. Ahmed, A. Awasthi, M.A. Stahlberg, A. Rajput, D.S. Magruder, . . . C. Dean. 2017. A novel method for culturing stellate astrocytes reveals spatially distinct Ca²⁺ signaling and vesicle recycling in astrocytic processes. *J Gen Physiol.* 149:149-170.10.1085/jgp.201611607
- Wolfes, A.C., and C. Dean. 2018. Culturing In Vivo-like Murine Astrocytes Using the Fast, Simple, and Inexpensive AWESAM Protocol. *Journal of visualized experiments : JoVE.*10.3791/56092

REVIEWERS' COMMENTS

Reviewer #1 (Remarks to the Author):

The reviewer appreciates the careful and thorough revision of the manuscript and the detailed comments and additional experiments which have improved the manuscript. The minor points that remain are listed below. The new Figure 6 is indeed very exciting and adds significant information.

Figure 1: The authors moved previously published mouse model data to the supplementary figures and added two bar graphs to the analysis of the new inducible Orai1KO experiments. I would recommend to indicate Tg-release in Fig 1E (as one would expect that the SOCE Area is analyzed, which would be near zero) and also to include the AUC of SOCE as a bar graph. In addition, Fig. 1F is confusing: Why is the relative Orai1 expression not reduced in the Cre+ tam+ induced cells?

Figure 1B is not mentioned in the text on page 7 and line 164, page 7 is confusing as the Tg peak is analyzed and the mRNA expression not altered. Please check figures versus text.

Analysis of expression of inflammasome related genes was carefully carried out and presented in the comments to reviewers (Fig. 1). Is there a mention of these important results within the manuscript? P2RX7 channels are not mentioned in the text, despite extensive literature on their role in inflammation, also in astrocytes. Please cite Zhao YF et al., 2022, for role of P2RX7 in depressive like behavior and add to discussion.

Same applies to the replies to additional points that yielded important and interesting results. The reviewer is aware of the already extensive supplementary results and figures, but the manuscript would benefit from addition of Figures 3 and 5 to the supplements and to the discussion. The dual activation protocol entails pretreatment with LPS (Signal 1) for several hours and then addition of ATP. How were the experiments in Reviewer Fig. 3 performed?

Reviewer #3 (Remarks to the Author):

Regulation of astrocyte-mediated brain inflammation by Orai1 channels.
Novakovic et al.

Astrocytes are a major CNS cell type and are increasingly recognized as important components of the CNS response to injury and disease, adopting what is commonly referred to as a 'reactive' phenotype. However, the molecular mechanisms underlying reactive astrogliosis are largely unknown.

Ca²⁺ release from internal stores, in an IP₃ dependent manner, is thought to be central to astrocyte function. Following depletion of internal Ca²⁺ stores, ongoing signaling must be maintained by store-operated Ca²⁺ entry (SOCE). Novakovic and colleagues have investigated the role of SOCE in regulating astrocyte reactivity by genetically ablating a key subunit of the Ca²⁺ release activated Ca²⁺ channel (CRAC) system, Orai1. Novakovic and colleagues report that deletion of Orai1 from astrocytes downregulates the expression of key glycolytic enzymes, metabolic intermediates and impairs ATP production. Furthermore, in their hands, Orai1 deletion reduces cytokine production in the hippocampus, reduces reactive astrogliosis and impacts inhibitory synaptic transmission in hippocampal CA1. Building on these observations, Novakovic et al., then report that mice with astrocyte-specific Ora1a are protected against inflammation induced depression.

Given the increasing recognition of astrocytes as key players in CNS disease, the manuscript is timely and the central theme of the manuscript appears novel.

This is a revision of a previously submitted manuscript.

Major issues:

In general, the authors answered all comments raised in review.

However, while I acknowledge the effort put into the new Ca²⁺ imaging data, I do not feel as though they fully answer the question of what is the mechanistic connection of Orai1-mediated Ca²⁺ signaling and the observed effects on synaptic transmission. At least one plausible explanation put forward by the authors is an effect on Ca²⁺-evoked gliotransmitter release. As the evidence points towards the gliotransmitter involved being ATP it would be relatively easy to pharmacologically test the link between Ca²⁺ signaling, ATP release and synaptic transmission.

The authors perform a comprehensive analysis of Ca²⁺ signaling in defined regions of the astrocytes (soma and processes). However, no mention of the complex structure of astrocytes was brought forward in the 'Introduction', nor the complex Ca²⁺ dynamics seen in these cells (despite this being suggested in the previous review). Furthermore, as Orai1 is thought to be involved in store-operated (ER) Ca²⁺ signaling in astrocytes, it would be helpful if the authors discuss what is known about ER extension into the processes and other possible modes of Ca²⁺ signaling (Semyanov et al., Nat Rev Neurosci: and references therein). Finally, the terminology 'distal processes' and 'proximal processes' is largely outdated and should be updated: ROIs for image analysis should also be included on Figure 6.

Reviewer #1

The reviewer appreciates the careful and thorough revision of the manuscript and the detailed comments and additional experiments which have improved the manuscript. The minor points that remain are listed below. The new Figure 6 is indeed very exciting and adds significant information.

Figure 1: The authors moved previously published mouse model data to the supplementary figures and added two bar graphs to the analysis of the new inducible Orai1KO experiments. I would recommend to indicate Tg-release in Fig 1E (as one would expect that the SOCE Area is analyzed, which would be near zero) and also to include the AUC of SOCE as a bar graph. In addition, Fig. 1F is confusing: Why is the relative Orai1 expression not reduced in the Cre+ tam+ induced cells?

As suggested by the reviewer, we have added panels showing area under the curve (AUC) of SOCE and of store-release (area under the thapsigargin part of the trace) as bar graphs. These are shown as panels F and G. The previous panel Figure 1F showed Orai1 expression in microglia (not astrocytes) to demonstrate the specificity of Orai1 deletion to astrocytes. This panel is now clearly labelled as microglia and moved next to the astrocyte mRNA panel (and relabeled as Figure 1C) for improved clarity.

Figure 1B is not mentioned in the text on page 7 and line 164, page 7 is confusing as the Tg peak is analyzed and the mRNA expression not altered. Please check figures versus text.

Thank you for catching this error. We have fixed the reference to Figure 1B and rewritten the text in line 164 for accuracy in citing the correct panels.

Analysis of expression of inflammasome related genes was carefully carried out and presented in the comments to reviewers (Fig. 1). Is there a mention of these important results within the manuscript? P2RX7 channels are not mentioned in the text, despite extensive literature on their role in inflammation, also in astrocytes. Please cite Zhao YF et al., 2022, for role of P2RX7 in depressive like behavior and add to discussion.

Thank you for the suggestion. We have added the inflammasome related gene expression to the Supplementary Figure 8 and discussed the results. Zhao et al 2022 is also cited now and discussed.

Same applies to the replies to additional points that yielded important and interesting results. The reviewer is aware of the already extensive supplementary results and figures, but the manuscript would benefit from addition of Figures 3 and 5 to the supplements and to the discussion. The dual activation protocol entails pretreatment with LPS (Signal 1) for several hours and then addition of ATP. How were the experiments in Reviewer Fig. 3 performed?

Thank you for the suggestion. We have added the LPS data to Supplementary Figure 7A and Supplementary Figure 11. In the experiment with LPS, we added both ATP and LPS at the same time as several protocols in the literature have employed the co-administration protocol.

Reviewer #3

Regulation of astrocyte-mediated brain inflammation by Orai1 channels. Novakovic et al.
Given the increasing recognition of astrocytes as key players in CNS disease, the manuscript is timely and the central theme of the manuscript appears novel.

This is a revision of a previously submitted manuscript.

Major issues: In general, the authors answered all comments raised in review.

However, while I acknowledge the effort put into the new Ca^{2+} imaging data, I do not feel as though they fully answer the question of what is the mechanistic connection of Orai1-mediated Ca^{2+} signaling and the observed effects on synaptic transmission. At least one plausible explanation put forward by the authors is an effect on Ca^{2+} -evoked gliotransmitter release. As the evidence points towards the gliotransmitter involved being ATP it would be relatively easy to pharmacologically test the link between Ca^{2+} signaling, ATP release and synaptic transmission.

We thank the reviewer for raising this point. To better explain and establish the connection between astrocyte Orai1 signaling and synaptic transmission, we have substantially rewritten the Discussion sections on pages 26-27 and the Results section on page 20. We hope that the new narrative will more clearly and directly explain the mechanistic basis of the reduction in inhibitory synaptic transmission observed in the conditional Orai1 KO mice.

In terms of a pharmacological experiment showing inhibition of ATP receptors and effects on synaptic transmission, in fact this experiment was already done previously (in our Toth et al, Science Signaling 2019 paper). In that study, we showed that applying a broad-spectrum purinergic receptor blocker (PPADS) at a dose that would inhibit P2Y receptors blocked the ATP-mediated enhancement of inhibitory synaptic transmission. Because the pharmacological experiment was already carried out in the exact same preparation and is published, we have not repeated it here.

The authors perform a comprehensive analysis of Ca^{2+} signaling in defined regions of the astrocytes (soma and processes). However, no mention of the complex structure of astrocytes was brought forward in the 'Introduction', nor the complex Ca^{2+} dynamics seen in these cells (despite this being suggested in the previous review). Furthermore, as Orai1 is thought to be involved in store-operated (ER) Ca^{2+} signaling in astrocytes, it would be helpful if the authors discuss what is known about ER extension into the processes and other possible modes of Ca^{2+} signaling (Semyanov et al., Nat Rev Neurosci: and references therein). Finally, the terminology 'distal processes' and 'proximal processes' is largely outdated and should be updated: ROIs for image analysis should also be included on Figure 6.

Thank you for this point. In the section that deals with analysis of the Ca^{2+} signals (page 17), we do point out that astrocyte morphology and structure is highly complex and shows dynamic Ca^{2+} signaling. We have not analyzed ER extension in different astrocyte compartments, and hence we would prefer not to get into ER morphology in this study, which deals more with a global analysis of Orai1 contributions to astrocyte Ca^{2+} signaling and not its cell biology. As our paper already contains a large amount of data on many aspects of astrocyte biology, we hope that an in-depth review of the ER cell biology is outside the scope of the paper. However, as suggested, we have added images in Supplementary Figure 11A,C showing representative ROIs in which the Ca^{2+}

traces were analyzed in the three compartments. Finally, we have chosen to retain the terminology for soma, proximal processes and distal processes as these are widely used in the field for astrocytic compartments.